# An Air Quality and Boundary Layer Dynamics Analysis of the Los Angeles Basin Area During the Southwest Urban NOx and VOCs Experiment (SUNVEx)

Edward J. Strobach[1,2], Sunil Baidar[1,2], Brian J. Carroll[1,2], Steven S. Brown[1,3], Kristen Zuraski[1,2], Matthew Coggon[1], Chelsea E. Stockwell[1], Lu Xu[4,5], Yelena L. Pichugina[1,2], W. Alan Brewer[1], Carsten Warneke[1], Jeff Peischl[1,2], Jessica Gilman[1], Brandi McCarty[1,2], Maxwell Holloway[1,2], and Richard Marchbanks[1]

[1]NOAA Chemical Sciences Laboratory, Boulder, CO 80309, USA
[2]Cooperative Institute for Research in Environmental Sciences, University of Colorado Boulder, Boulder, CO 80309, USA
[3]Department of Chemistry, University of Colorado Boulder, Boulder, CO 80309, USA
[4]Department of Energy, St. Louis, MO 63130, USA
[5]Department of Environmental and Chemical Engineering, Washington University at St. Louis, St. Louis, MO 63130, USA

**Correspondence:** Edward J. Strobach (edward.strobach@noaa.gov)

**Abstract.** The NOAA Chemical Sciences Laboratory (CSL) conducted the Southwest Urban $NO_x$ and VOCs Experiment (SUNVEx) to study emissions and the role of boundary layer (BL) dynamics and seabreeze (SB) transitions on the evolution of coastal air quality. The study presented utilizes remote sensing and in situ observations in Pasadena, California. Separate analyses are conducted on the synoptic conditions during ozone ($O_3$) exceedance (>70 ppb) and non-exceedance (<70 ppb) days, and the fine structure variability of in situ chemistry measurements during BL growth and SB transitions.

Diurnal analyses spanning August 2021 revealed a markedly different wind direction during evenings preceding $O_3$ exceedance (northerly) versus non-exceedance (easterly) days. Increased $O_3$ occurred simultaneously with warmer and drier conditions, a reduction in winds, and an increase volatile organic compounds (VOCs) and fine particulate matter ($PM_{2.5}$). While the average BL height was lower and surface pressure was higher, the day-to-day variability of these quantities led to an overall weak statistical relationship. Investigations focused on the fine structure variability of in situ chemistry measurements superimposed on background trends were conducted using a novel Multivariate Spectral Coherence Mapping (MSCM) technique that combined the spectral structure of two or more independent measurements through a wavelet analysis as reported by maximum-normalized scaleograms. A case study was chosen to illustrate the MSCM technique, where the dominant peaks in scaleograms were identified and compared to BL height during the growth phase. The temporal widths of peaks ($\tau_{max}$) derived from VOC and nitrogen oxide ($NO_x$) scaleograms, and scaleograms combining VOCs, $NO_x$, and variations in BL height indicated a broadening with respect to time time as the BL increased in depth. A separate section focused on comparisons between $\tau_{max}$ and BL height during August 2021 revealed uncorrelated or weakly correlated scatter, except in the case of VOCs when really large $\tau_{max}$ and relatively deep BL heights were ignored. Instances of large $\tau_{max}$ and relatively deep BL heights occurred near sunrise and as onshore flow entered Pasadena, respectively. Wind transitions likely influenced both the dynamical evolution of the BL and tracer advection, and thus offer additional challenges when separating factors contributing

to the fine structure. Other insights gained from this work include observations of descending wind jets from the San Gabriel Mountains that were not resolved by the HRRR model, and the derivation of intrinsic properties of oscillations observed in $NO_x$ and $O_3$ during the interaction between a SB and enhanced winds above the BL that flowed in opposition to the SB.

## 1 Introduction

Understanding the regional air quality and meteorological conditions in the Los Angeles (LA) basin has been a research interest for decades as a result of historically high pollution levels that have affected the area (Warneke et al., 2012). Reductions in precursor emissions has led to a declining trend in ozone ($O_3$) over the period extending from the 1960s up to the 21$^{st}$ century (Parrish et al., 2016). Since 2010, however, $O_3$ levels have not decreased further, despite continued declines in precursor emissions as evident in the annual maximum 8-hr average $O_3$ hovering around 100 ppb.

Most recent efforts in isolating the emissions responsible for the secondary formation of $O_3$ in the LA basin have found a wide range of emissions that include vehicles, volatile chemical products and biogenics as sources of $O_3$ precursors. Ryerson et al. (2013) found that carbon monoxide (CO), nitrogen oxides ($NO_x$), and volatile organic compounds (VOCs) were dominated by vehicular emissions. Gu et al. (2021) found that the top 10 VOCs were strongly influenced by traffic. Comparisons between weekend versus weekday emissions highlighted the important role of $NO_x$ on $O_3$ formation since most of the "heavy-

duty trucking" occurred on weekdays (Nussbaumer and Cohen, 2020). The isolation of alkenes from VOC measurements has highlighted motor vehicles as being an important source while at the same time underscoring the complex relationship between alkenes and hydroxyl radicals (OH) (de Gouw et al., 2017; Hansen et al., 2021). Hasheminassab et al. (2014) determined that vehicle emissions were the second most important source to $PM_{2.5}$, slightly behind secondary nitrates. However, a study in 2018 found that sources other than vehicular emissions related to pesticides, cleaning products, and personal care products (i.e.,

volatile chemical products–VCPs) contributed to twice the amount of VOCs compared to vehicular emissions, thus pointing to a recent shift in the dominant emission sources contributing to $O_3$ precursors (McDonald et al., 2018).

    Other studies have highlighted different factors responsible for poor air quality. Gu et al. (2021) demonstrated that "greening" a city can increase biogenic VOCs, which could have unintended consequences for $O_3$ production. Muñiz-Unamunzaga et al. (2018) determined that halogen and sulfer-based compounds can modify $O_3$ and $NO_x$ concentrations in coastal environments

spurred by changes in OH chemistry and the $HO_x$ (or $RO_x$) cycle. Nussbaumer and Cohen (2021) noted that warmer days (and nights) can lead to elevated VOC abundance, which often, but not always, coincides with conditions associated with high pressure ridging and stagnant condtions. Thus, in the case of the latter, understanding the broader meteorological conditions and the boundary layer (BL) evolution becomes important when addressing changes in atmospheric chemistry and air quality.

    Composite analyses of different types of large-scale patterns have often shown stagnant high pressure as being an ideal

condition for poor air quality (e.g., Lai and Cheng, 2009; Zhou et al., 2018; Nauth et al., 2023). However, the study by Nauth et al. as well as others (e.g., Peterson et al., 2019; Wang et al., 2019) have shown that interactions between different air masses, blocking patterns, and the pressure pattern arrangement (regardless of high or low pressure) can also lead to poor air quality episodes. While the latter two are larger-scale in nature, the interaction between air masses can be large-scale, mesoscale,

or both. For instance, Nauth et al. (2023) found that synoptic northwesterly flows and the simultaneous development of a seabreeze (SB) from the south enhanced $O_3$ as the two flows merged to form a convergence line over the New York city area. Their results confirm findings from previous studies examining locally generated sea and bay breezes (e.g., Banta et al., 2005; Loughner et al., 2014). In the LA basin, modeling and observational studies have highlighted the role of mesoscale transport from SBs penetrating inland, resulting in a more complex set of chemical reactions that stem from the mixing of air between marine and terrestrial BLs (Lu and Turco, 1995; Wagner et al., 2012).

In addition to SBs generated by strong thermal contrasts between the LA basin and the coastal ocean are impacts from the complex topography to the north and east that alter the flow field as differential heating across terrain slopes generate upslope and downslope winds that can modify or interact with a developing SB (Pérez et al., 2020). Langford et al. (2010) found that upslope flow within the BL and forcing conditions favoring westerly flow above the BL can lead to significant transport out of the LA basin into the free troposphere through a phenomenon known as BL venting (Loughner et al., 2014). However, it is not clear how common the case described by Langford et al. is, and what combination of forcing conditions associated with SB development, background synoptic pressure gradient, and topographically driven flows are required to prevent transport out of the LA basin and subsequently poor air quality conditions. Thus, although we may know the factors responsible for mesoscale and large-scale forcing, we do not necessarily know how different wind regimes will interact and modify the BL dynamics. Furthermore, local impacts from urban development adds to the dynamical complexity in the form of the well-documented urban heat island (UHI) effect, which can modify SB propagation (Yoon-Hee et al., 2016) while at the same time alter the wind profile structure in the form of the urban wind island (UWI) effect (Droste et al., 2018; Baidar et al., 2020).

To address questions related to the role of dynamics on air quality in the LA basin, the NOAA Chemical Sciences Laboratory (CSL) deployed instrument payloads during August 2021 in an effort known as the Southwest Urban $NO_x$ and VOCs Experiment (SUNVEx). Instruments featuring in situ chemistry/meteorology and a Stationary Doppler lidar On a Trailer (StaDOT) were stationed in Pasadena, CA, while a mobile component was deployed to survey the regional air quality and BL dynamics. Here, we focus on data collected at Pasadena and evaluate the broader conditions observed during the month of August as well as an examination of BL growth and SB transitions on the finer scale features observed in air quality measurements. The close proximity of stationary systems in Pasadena allowed the characterization of local temporal changes in air quality and dynamics observations with the aid of a wavelet technique to determine the fine structure characteristics associated with BL dynamics within variable time series, while a mapping technique was developed to quantitatively examine the variability between independent measurements to understand dynamical linkages to air quality evolution. Results highlighting BL transitions and the role of BL growth, in particular, have major implications in air quality modeling since it is those situations where models tend to struggle (Sastre et al., 2015), and represents a key area that the authors aim to address as part of this work. Furthermore, the techniques developed in this study allow a quantitative analysis of the fine structure variability of air quality and BL dynamics observations as well as variable interdependencies during BL transitions, which, to the authors' knowledge, has never been done at this level of detail.

The remainder of the study is as follows. Section 2 describes the data used in the study and data processing methods. For data processing methods, we adopt a wavelet technique to isolate the local characteristics of the data time series after the removal

of the diel cycle using empirical mode decomposition (EMD) so that maximum normalized scaleograms of multiple variables can be compared using a method that we call the Multivariate Spectral Coherence Mapping (MSCM) technique. Section 3 presents an analysis of the large-scale changes in air quality and dynamics measurements spanning the month of August. A case study is chosen and discussed in Section 4 that describes the dynamical evolution that took place on 16 August 2021, and the role that BL transitions and SB development had on air quality evolution. Section 5 expands upon analyses conducted in Section 4 by analyzing the impact of BL growth on the temporal variations in air quality and dynamics measurements during August. Conclusions with a description of study limitations and a path forward is left for Section 6. Appendices A-D describe an algorithmic technique for spectral mapping scaleograms through a variable ranking approach, an approach to modeling the BL height during the growth phase, an analysis of descending winds above the BL for the case study analyzed, and the intrinsic features of the oscillations observed in measurements during the passage of a SB.

## 2  Data and Methods

Observations in Pasadena, CA featuring DL and in situ chemistry/meteorology payloads are presented in the Data section along with details of the High Resolution Rapid Refresh (HRRR) model used to describe the regional conditions. Other data used include the $O_3$ measurements from the AIRNOW network (https://www.AIRNOW.gov/) overlaid against output from the HRRR.

A methods section is dedicated to the application of a Ricker wavelet to stationary measurements after the removal of the time varying mean signal and the development of the Multivariate Spectral Coherence Mapping (MSCM) technique to compare the temporal likeness between multiple variables.

### 2.1  Data

#### 2.1.1  Stationary Doppler lidar On a Trailer (StaDOT)

The StaDOT is a stationary DL that was deployed in Pasadena, CA (Stationary Doppler lidar On a Trailer–StaDOT) to measure winds spanning 05 August 2021 through 02 September 2021. Wind profile measurements (direction and speed) were derived from conical scans at 15, 35, and 60-degree elevation angles. The shallower 15-degree elevation angle resulted in higher resolution winds closer to the surface of about 20 m that decreased to 67 m at a height of 6000 m as the elevation angle increased. The time it took to perform conical scans was 3.5 minutes, with each revolution resulting in 120 azimuthal angles and a line of sight (LOS) velocity measurement at a 2 Hz resolution.

Following conical scans were vertical stares to measure vertical winds over a 11.5-minute period, which were used to derive vertical velocity variance, skewness, and kurtosis. Other products that were available included the signal-to-noise (SNR) ratio and the derivation of BL heights using the fuzzy logic approach from Bonin et al. (2018). The total time to complete a scan cycle was 15 minutes, thus representing the time resolution between horizontal and vertical wind measurements. Additional

details related to the capability of the instrument as well as recently developed techniques can be found in Schroeder et al. (2020) and Strobach et al. (2023).

In this study, the horizontal and vertical winds, and BL height from the DL are used to describe the evolving conditions at Pasadena spanning nocturnal and daytime periods during the month of August. Additionally, departures in the mean wind and BL structure as observed by the DL are examined when addressing the fine-scale variability reported in scaleograms (discussed in the Methods subsection).

### 2.1.2 $NO_2$, $NO_x$, $O_x$, and Meteorology Measurements

In situ measurements of $NO_x$, $NO_y$, and $O_x$ were conducted at the Caltech campus in Pasadena, CA using a 10-meter inlet tower. Three different instruments were employed, including a cavity ringdown spectroscopy (CRDS) instrument, a laser-induced fluorescent (LIF) instrument, and a commercial $O_3$ analyzer (TECO, Thermo Environmental Instruments, model 49c). The CRDS instrument, which has been described previously (Fuchs et al., 2009; Rollins et al., 2020; Washenfelder et al., 2011; Wild et al., 2017), consisted of four channels. $NO_2$, $NO_x$, $O_x$, and $NO_y$ measurements were observed by directly measuring $NO_2$ using a 405 nm laser, following chemical and thermal conversions for the $NO_x$, $O_x$ and $NO_y$ species. Excess $O_3$ was introduced to ambient NO to convert it to $NO_2$ for the $NO_x$ measurement, while excess NO was added to ambient $O_3$ to convert it to $NO_2$ for the $O_x$ measurement. A heated inlet (T = 650 °C) was employed to thermally dissociate $NO_y$ to NO and $NO_2$, where excess $O_3$ was again added to convert NO to $NO_2$ for the $NO_y$ measurement. Data was collected at 1 Hz, and the accuracy of the measurements ranged from 3-5% for NO, $NO_2$, and $NO_x$, and 12% for $NO_y$. The LIF instrument, previously described by Rollins et al. (2020), directly measured NO at 1 Hz with an uncertainty of 6-9% and a limit of detection of 1 ppt. This two-channel instrument converted $NO_2$ to NO using a blue light converter for the $NO_x$ measurement.

The data obtained from the three instruments were consolidated into a merged file, accessible online (Brown, 2021). The combined datasets prioritized the most direct instrument measurements when available. For $NO_2$, the CRDS data took precedence, with data from the LIF instrument filling in any dataset gaps. In the case of the $NO_x$ data, NO data from LIF combined with $NO_2$ from the CRDS instrument was used when both were available, the CRDS data was used when the LIF instrument was inactive, and vice versa. $O_3$ data primarily came from the TECO instrument, except for the early campaign period when the CRDS $O_3$ data was used. $NO_y$ was exclusively measured by the CRDS instrument, and this dataset remained independent of the other instruments. The uncertainty for each species depended on the instrument that measured the data and is also reported in the merge file. For the $NO_x$ data, where NO from the LIF instrument combined with $NO_2$ from the CRDS instrument, propagation of error was employed to represent the uncertainty in both measurements.

A separate instrument payload to measure in situ meteorology was installed at CalTech. Measurements included temperature, pressure, relative humidity, wind speed, and wind direction recorded at a 1-minute time resolution. In this study we consider only temperature, pressure, and relative humidity since we rely on winds from the DL.

### 2.1.3 VOC Measurements

VOC mixing ratios were monitored using a Vocus proton-transfer-reaction time-of-flight mass spectrometer (PTR-ToF-MS, Krechmer et al., 2018). The PTR-ToF-MS was operated as described by Coggon et al. (2023). Briefly, ambient air was sampled through a ≈1 meter Teflon tube and VOC mixing ratios were measured at 1 Hz. Instrument background were determined every 2 hr by sampling a platinum catalyst heated to 350 °C. Mixing ratios were determined for small oxygenates (ethanol, methanol, acetone, acetaldehyde, methyl vinyl ketone + methacrolein), C6-C9 aromatics, biogenic VOCs (isoprene, monoterpenes), and nitriles (acetonitrile, benzonitrile) using gravimetrically-prepared standards. These VOCs have reported uncertainties of 20%. Sensitivities for other masses reported by the PTR-ToF-MS were estimated by the methods described by Sekimoto et al. (2017) and have uncertainties greater than 50%. Here, we report total VOCs as the sum of PTR-ToF-MS mixing ratios, which is used in analyses later in the manuscript.

### 2.1.4 High Resolution Rapid Refresh

To describe the meteorological conditions when evaluating the 16 August case study, we use the hourly output from version 4 (v4) of the 3 km High-Resolution Rapid Refresh (HRRR) model. HRRRv4 includes notable improvements to the Mellor-Yamada-Nakashini-Niino (MYNN) BL scheme related to subgrid-scale (SGS) clouds, a 36-member ensemble used for data assimilation to address uncertainty in the "initial conditions and model physics" to improve representativeness of complex flow environments, the inclusion of radar observations to improve the representation of clouds, predictions of wildfire smoke transport, and modifications to radiative transfer from SGS clouds. Also included is the 9 soil-layer Rapid Update Cycle (RUC) land surface model and the aerosol-aware Thomson-Eidhammer microphysics scheme. More details related to HRRRv4 can be found in Dowell et al. (2022).

When evaluating the regional conditions, we defined a plotting domain encompassing the LA basin, topography to the north and east, and the coastal Pacific Ocean along the shoreline defining one of the geographical borders outlining the LA basin. The broader synoptic conditions are also described with links to synoptic maps.

## 2.2 Methods

### 2.2.1 Empirical Mode Decomposition (EMD)

A time varying signal represented as a superposition of scales containing localized changes in the frequency structure can be decomposed via empirical mode decomposition (EMD) and represented as distinct intrinsic mode functions (IMFs) of a finite number. Pioneered by Huang and Wu (2008) to study nonlinear waves, the decomposition into IMFs is an integral component of the Hilbert-Huang Transform (HHT) to address the nonstationarity and nonlinearity of a signal. In this study, the EMD portion of the HHT is used to separate observables into IMFs and a residual.

There are two requirements when constructing IMFs: 1) the number of extrema and zero crossings must equal or differ at most by one, and 2) the mean of the envelope tracing local minima and maxima is identically zero (an example of this

procedure can be found in Figure 2 from Huang and Wu (2008)). Incorporating these requirements as conditions into an algorithm through an iterative "sifting" approach enables the separation of $N$ IMFs whose spectral characteristics range from a high-to-low frequency structure. The high frequency structure is isolated first in the sifting process and removed before the next IMF is determined. Since the higher frequency content has been separated, then the subsequent IMFs will contain lower frequency content. Eq. (1) represents the decomposition of an arbitrary signal, $x(t)$, into a summation of IMFs ($x_i'(t)$) and a residual ($x_r(t)$)

$$x(t) = x_r(t) + \sum_{i=1}^{N} x_i'(t) \tag{1}$$

The residual represents the portion of the signal that is either constant or features a time varying mean that did not satisfy one of the two requirements above. The number of modes varies depending on the data collected, the type of observations or variables being processed, or the resolution of measurements, and can be determined by the limit that $x_i'(t)$ goes to zero, i.e.,

$$\lim_{x_i'(t) \to 0 \ \forall \ t} i = N + 1 \tag{2}$$

The number of IMFs, $N$, is truncated according to Eq. (2), where the summation of IMFs is used in Sections 4 and 5 to isolate variations due to the diel cycle from the fine structure variability. Thus, we use the summation of IMFs defined by Eq. (3) when conducting a wavelet analysis in frequency space and when applying the Multivariate Spectral Coherence Mapping (MSCM) technique that is described later.

$$\tilde{x}(t) = \sum_{i=1}^{N} x_i'(t) = x(t) - x_r(t) \tag{3}$$

Figure 1a shows a 24-hour period of measured $O_3$ (blue) and the residual (red). The residual represents the diurnal structure of $O_3$, while Figure 1b represents the summation of IMFs as defined by Eq. (3). The positive and negative values in $\tilde{O}_3$ represent the fine scale variability superimposed on the diurnal trend.

### 2.2.2 The Ricker Wavelet and the scaleogram

In order to analyze the fine structure variability of chemistry and meteorological measurements to understand the role of BL transitions, we apply a wavelet across 24-hour blocks of data spanning the diel cycle. For this analysis we chose the continuous Ricker wavelet defined by Eq. (4)

$$\psi\left(\frac{t-b}{\tau}\right) = \frac{2}{\sqrt{3\tau}\pi^{1/4}}\left(1 - \left(\frac{t-b}{\tau}\right)^2\right) e^{\frac{-(t-b)^2}{2\tau^2}} \tag{4}$$

where $\tau$ is the width or dilation of the wavelet, and $b$ is the position of the wavelet along the data time series. The wavelet is symmetrical and is derived from normalizing the negative of the second derivative of a Gaussian function with respect to time, thus leading to a structure that resembles a sombrero.

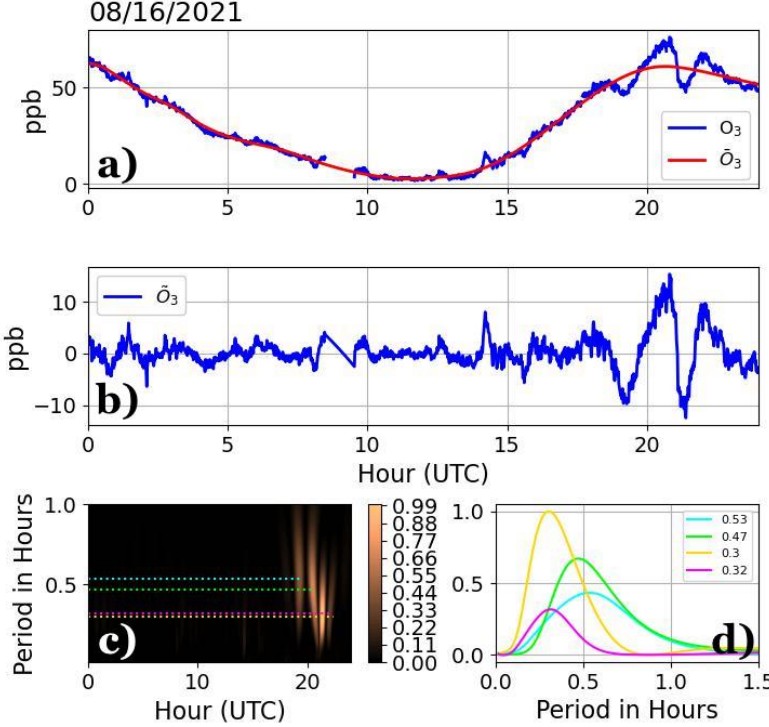

**Figure 1.** a) Measurements of $O_3$ (blue) overlaid with the portion of measurements driven by the diel cycle (red), b) the variability of $O_3$ after the removal of the time varying mean, c) a scaleogram using results from b) with different wavelet dilations overlaid with dotted lines highlighting peaks in the scaleogram, and d) the maximum normalized power spectrum density (PSD) color-matched with dotted lines in c). The legend in d) represents the temporal width associated with the dominant extremum in the PSD.

The summation of IMFs in Eq. (3) is convolved with Eq. (4) to obtain a power spectrum, $\widetilde{W}_\psi(\tau, b)$, at a given dilation, $\tau$, via Eq. (5)

$$\widetilde{W}_\psi(\tau, b) = \int_{-\infty}^{\infty} \tilde{x}(t)\psi\left(\frac{t-b}{\tau}\right)dt \qquad (5)$$

Increasing or decreasing the dilation leads to isolating the slow or rapidly varying portions of the signal, respectively, which when grouped together produces a 2D spectrum dependent on dilation ($\tau$) and the position of the wavelet ($b$), the result of which is joined together to produce a scaleogram as shown in Figure 1c. The scaleogram represents the maximum normalized power spectrum density (PSD) with respect to time as the wavelet is translated across the time series (b–x-axis), and with

215 respect to wavelet dilation, $\tau$ (dilation–y-axis), to isolate localized data spikes that feature different temporal widths, where the

maximum normalized PSD is simply represented as

$$\widetilde{\chi}_\psi(\tau, b) = \frac{\widetilde{W}_\psi(\tau, b)}{max(\widetilde{W}_\psi(\tau, b))} \tag{6}$$

### 2.2.3 Isolating Peaks in the Scaleogram

Since each time within a scaleogram corresponds to a series of real-valued coefficients derived from changing the wavelet dilation, then it is possible to determine the temporal width (or dilation) associated with a maximum PSD at each time step (i.e., $max(\widetilde{\chi}_\psi(\tau, b)) = \widetilde{\chi}_\psi(\tau_{max}, b) \rightarrow \widetilde{\chi}_\psi(b)$). Once the maximum PSD is found at each time step, then the time-ordered output can be sorted from increasing to decreasing maximum PSD (i.e., $\widetilde{\chi}_\psi(b) \rightarrow \widetilde{R}_\psi(k) = \widetilde{R}_\psi^k$) to isolate the peaks that stand out from the rest of the dataset, where $k$ replaces $b$ as the time-independent sorted index. By definition, the sorted output for a single variable being processed through a wavelet transform and normalized by the maximum PSD will begin with a value of 1 according to Eq. (6). From there, the output of the sorted array decreases toward zero. We identify significant peaks in $\widetilde{R}_\psi^k$ by comparing neighboring peaks via Eq. (7),

$$\Gamma = \sum_{\substack{k \\ \Gamma > 0.5 \vee \widetilde{R}_\psi^{k+1} \geq 0.25}} \frac{\widetilde{R}_\psi^{k+1}}{\widetilde{R}_\psi^k} \tag{7}$$

where the conditions within the summation terminate the algorithm if the subsequent peak within the sorted array is less than half the value of the larger adjacent peak, or if $\widetilde{R}_\psi^{k+1}$ is less than or equal to 0.25. The conditions were chosen based on trial and error, and are necessary if we are only interested in isolating clear signatures linked to micrometeorological dynamics.

Figure 1c includes overlays of dotted lines that identify the peaks determined as significant using the method outlined above, while Figure 1d shows the corresponding $\widetilde{\chi}_\psi(b)$ color-matched with dotted lines in c). The PSD associated with $\widetilde{\chi}_\psi(b)$ in Figure 1d resembles a gamma distribution-like character, with peaks in $\widetilde{\chi}_\psi(b)$ decreasing from 0.53 to 0.3 hours.

### 2.2.4 The Multivariate Spectral Coherence Mapping (MSCM) Technique

To compare the PSD between variables, we use the maximum normalized PSD defined by Eq. (6), which results in a range of values between 0 and 1 as shown in the color-scale in Figure 1c. This is necessary since we are comparing the spectral structure of variables with unlike units, and since the variability may not change proportionately. Furthermore, ensuring that the normalized power spectrum distribution falls between 0 and 1 allows the mapping of multiple power spectrum distributions onto a single scaleogram with values also falling between 0 and 1. For two variables, $\widetilde{\chi}_\psi^1$ and $\widetilde{\chi}_\psi^2$, we multiply and raise the product to the 1/2 to define a type of cross-spectrum between two sets of observations. In a general sense, we can extend the operation to an arbitrary number of variables with Eq. (8)

$$\widetilde{C}^L(\tau, b) = \Big( \prod_{j=1}^L \widetilde{\chi}_\psi^j(\tau, b) \Big)^{1/L} \tag{8}$$

where $L$ is an arbitrary number of variables and $j$ represents the variable number index. The superscript in $\widetilde{\chi}_\psi^j(\tau, b)$ does not represent a power, but rather a power spectrum associated with a variable assigned to index $j$. Eq. (8) is basically the geometric

mean of power spectra for $L$ variables normalized by their respective maximum power. Though not a measure of coherence in the traditional sense, which uses the formal cross-spectrum definition between variables after being processed through a Fourier transform, the mapping of normalized power spectra highlights temporal widths within the dataset where variables exhibit similar spectral variability, thus revealing instances between datasets where there is structural coherence. Furthermore, this method is slightly different than wavelet coherence methods (e.g., Grinsted et al., 2004) in that the maximum PSD is used over the standard deviation when normalizing, a smoothing operator is not used, and we extend the analysis to any number of variables, not just two.

Eq. (8) provides the advantage of being able to determine variables that vary together in time, thus potentially enabling the determination of conversion relationships between atmospheric compounds and the role of BL dynamics on time changes in chemistry measurements. We call this method the Multivariate Spectral Coherence Mapping (MSCM) technique since we are able to produce scaleograms for an arbitrary number of variables that vary in time. Additionally, we can order scaleograms according to spectral likeness with a reference variable. For instance, if we define a reference variable, $\widetilde{\chi}_\psi^{m_0}$, and compute all possible variable pairings (2 variables) with the remaining $M-1$ variables that are left, then we have a vector space defined by

$$\widetilde{C}^2 = \{(\widetilde{\chi}_\psi^{m_0}\widetilde{\chi}_\psi^{m_1})^{1/2}, (\widetilde{\chi}_\psi^{m_0}\widetilde{\chi}_\psi^{m_2})^{1/2}, ..., (\widetilde{\chi}_\psi^{m_0}\widetilde{\chi}_\psi^{m_{M-1}})^{1/2}\} \tag{9}$$

Each pairing within the vector space in Eq. (9) can be summed with respect to time and dilation using Eq. (10)

$$S_m = \sum_{i=1}\sum_{l=1}\widetilde{C}_m^2\left(\tau_i, b_l\right) \tag{10}$$

where $m$ represents the index associated with an arbitrary variable pairing in Eq. (9), $i$ is a dummy index for dilation, and $l$ is the time index. The $M-1$ summations of Eq. (9) via Eq. (10) can be used to sort the array of $S_m$'s and indices in descending order, i.e.,

$$max\left(\widetilde{C}^2\right) = \{(\widetilde{\chi}_\psi^{m_0}\widetilde{\chi}_\psi^{n_1})^{1/2}, (\widetilde{\chi}_\psi^{m_0}\widetilde{\chi}_\psi^{n_2})^{1/2}, ..., (\widetilde{\chi}_\psi^{m_0}\widetilde{\chi}_\psi^{n_{M-1}})^{1/2}\} \tag{11}$$

where $n$ has replaced $m$ in Eq. (9) as a result of being maximum sorted. The result above can be expressed in general terms for an arbitrary number of variable groupings. Eq. (11) is used to determine groupings of two (or more) variables that share the strongest spectral likeness using calculations in Eq. (10). Thus, the order of scaleograms presented in Figures 8 and 9 is based on a decreasing sum from a) to i). An appendix is included below that discusses the algorithmic procedure for optimal ordering of variables based on maximum pattern likeness of $M$ variables (Appendix A).

## 3   Results

Most of the measurements taken simultaneously by the DL and in situ instruments in Pasadena, CA occurred during August 2021. Figure 2 shows a) $O_3$, b) $NO_x/NO_y$, c) VOCs, d) $\sum_i VOC_i/NO_x$, e) $O_x$, and f) $PM_{2.5}$. Figure 3 shows a) temperature, b) relative humidity, c) BL-averaged wind speed, d) BL-averaged wind shear, e) BL height, and f) pressure. Both Figures 2

and 3 are overlaid with a 25 ppb contour of $NO_x$ and a 70 ppb contour of $O_3$ in magenta and gray, respectively. We chose an $O_3$ threshold of 70 ppb to define $O_3$ exceedance, realizing that the national air quality standard definition for $O_3$ exceedance applies to $O_3$ averaged over 8-hour periods rather than to instantaneous measurements. For chemistry observations, we included

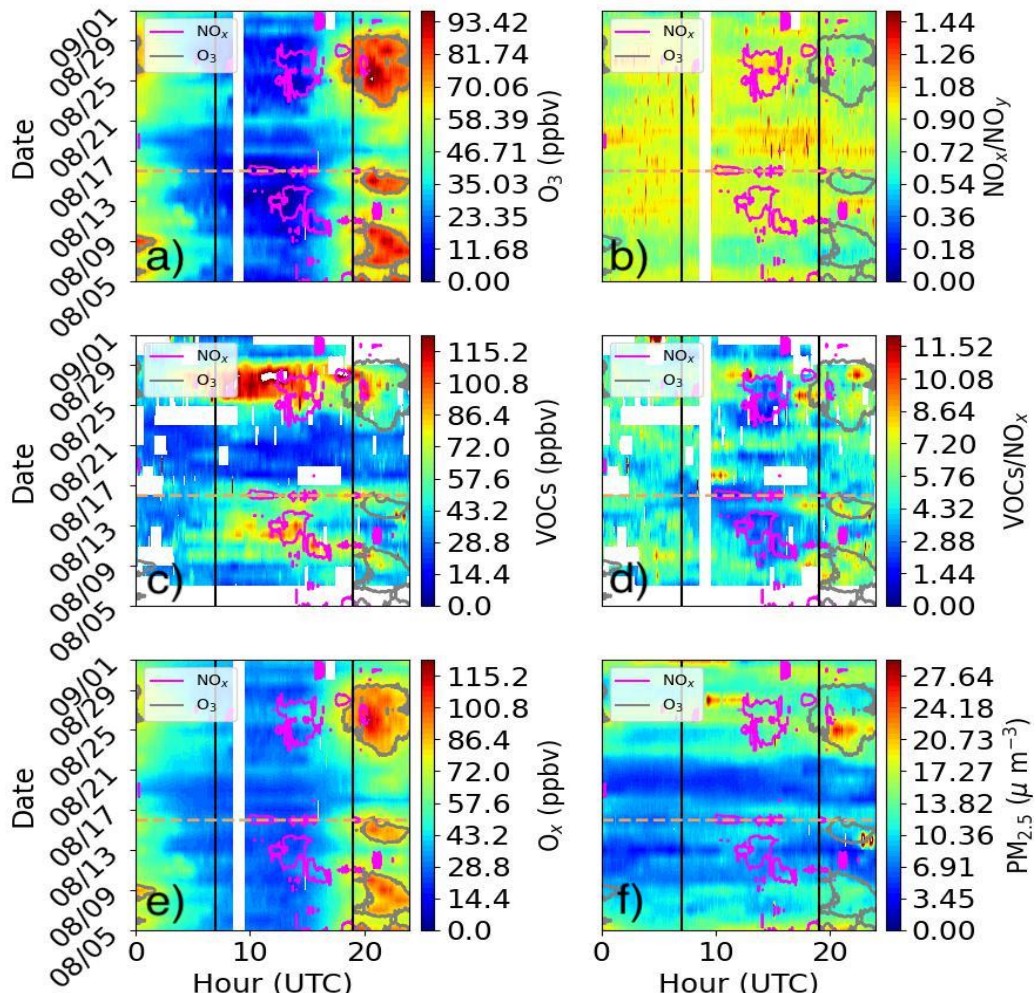

**Figure 2.** in situ observations of a) $O_3$, b) $NO_x/NO_y$, c) VOCs, d) $\sum_i$VOCs/$NO_x$, e) $O_x$, and f) $PM_{2.5}$ in Pasadena, CA overlaid with a 25 ppb $NO_x$ contour in magenta and a 70 ppb $O_3$ contour in gray. Included is a day chosen for a case study (gold dashed line). Vertical black lines denote midnight (7 UTC) and noon (19 UTC).

$NO_x$ and VOCs because of the well documented influence on $O_3$ production, $NO_x/NO_y$ to identify whether emissions were recent since it takes time for $NO_x$ to oxidize into other compounds (e.g., HONO, PAN, etc.), the ratio between total VOCs to
280 $NO_x$ to highlight $NO_x$-sensitive versus VOC-sensitive regimes, the summation of $O_3$ and $NO_2$ (i.e., $O_x$) to examine whether

interactions between $NO_2$ and $O_3$ were conserved, and $PM_{2.5}$ as an air quality metric. Meteorological measurements were chosen to examine the thermodynamic and dynamic characteristics under varying synoptic conditions.

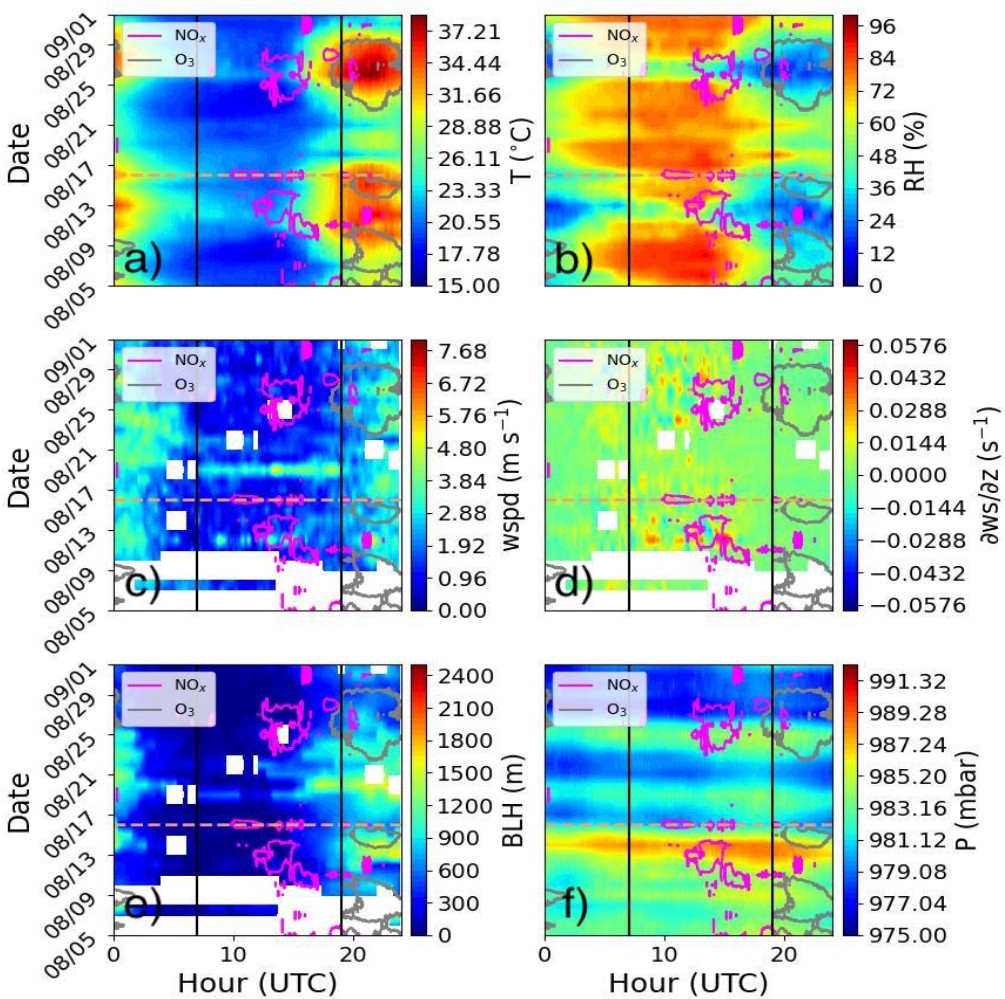

**Figure 3.** Observations of a) surface temperature, b) surface relative humidity, c) BL-averaged wind speed, d) BL-averaged wind shear, e) BL height. and f) surface pressure in Pasadena, CA overlaid with a 25 ppb $NO_x$ contour in magenta and a 70 ppb $O_3$ contour in gray. Included is a day chosen for a case study (gold dashed line). Vertical black lines denote midnight (7 UTC) and noon (19 UTC).

The diurnal structure of $O_3$ in Figure 2a is demonstrated by a minimum at night and a maximum during the day that closely resembles the diurnal temperature structure in Figure 3a. The morning transition that is driven by increased surface temperature
accelerates chemical reactions of most atmospheric compounds (Pusede et al., 2014) while increased downward solar short-

**Table 1.** Correlation coefficients between listed variables and $O_3$ for the month of August from 16 utc to 23:59 utc.

| | $NO_x/NO_y$ | $VOCs/NO_x$ | T | RH | BLH | P | VOCs | $PM_{2.5}$ |
|---|---|---|---|---|---|---|---|---|
| $\rho$ | -0.55 | 0.57 | 0.69 | -0.64 | 0.17 | -0.09 | 0.31 | 0.39 |

wave flux promotes photochemistry that leads to $O_3$ production. Periods where $O_3$ exceeded 70 ppb occurred in clusters during 08/05-08/11, 08/14-08/16, and 08/23-08/29 (17 of 28 days sampled in August), and occurred on both weekdays and weekends.

A strong overlap between periods of $O_3$ exceedance and increased (decreased) temperature (relative humidity) in Figure 3a(b) during the day, especially in the latter part of the month, is illustrated by the gray 70 ppb $O_3$ contour. A transition from deep to shallow daytime BL heights in Figure 3e and a slight reduction in BL-averaged wind speed in Figure 3c coincided with periods of $O_3$ exceedance. Other meteorological fields such as BL-averaged wind speed shear ($\partial ws/\partial z$–Figure 3d) and surface pressure (Figure 3f) did not exhibit clear contrasting patterns between $O_3$ exceedance and non-exceedance days. Figure 2b shows decreased $NO_x/NO_y$, particularly during $O_3$ exceedance days, that overlapped with increased $VOCs/NO_x$ ratios in excess of 5 as shown in Figure 2d. $VOCs/NO_x$ ratios in excess of 5 are known to favor increased $O_3$ production that can ultimately lead to $O_3$ exceedance as evidenced by the 70 ppb contour encapsulating relatively high $VOCs/NO_x$ ratios (Seinfeld and Pandis, 2016). The simultaneous decrease in $NO_x/NO_y$ and increase in $VOCs/NO_x$ is further exemplified by Figure 2c, which shows VOCs remaining elevated while $NO_x$ significantly depleted into the day. The depletion of $NO_x$ and increase in VOCs into the day coincided with large increases in $O_3$ and $O_x$ in Figure 2e. Instances of elevated VOCs, and thus elevated $O_3$, occurred during days that were more polluted overall as indicated by $PM_{2.5}$ in Figure 2f. Some studies have shown VOC concentrations increase with temperature (e.g., Nussbaumer and Cohen, 2021), which may partially explain the longevity of VOCs well into the afternoon despite convective BL mixing. Table 1 summarizes the correlation coefficients for key chemistry and meteorological variables discussed. As can be seen, higher correlations (and anti-correlations) are found in temperature, relative humidity, VOCs, $PM_{2.5}$, $NO_x/NO_y$, and $VOCs/NO_x$ when compared to $O_3$. Variables that are uncorrelated with $O_3$ are surface pressure and BL height. As noted earlier, $O_3$ exceedance periods straddled transitions from high-to-low BL height as well as surface pressure during a limited sampling period of about a month.

Studying the evening conditions leading up to $O_3$ exceedance days is also very important. Figure 2a shows relatively lower $O_3$ concentrations during evenings that preceded $O_3$ exceedance days after 12 August. Furthermore, evenings with relatively low $O_3$ coincided with increased $NO_x$ as a result of increased $NO_2$ and NO (magenta contour in Figure 2a), increased $NO_x/NO_y$ (Figure 2b–regardless of whether or not an evening preceded an $O_3$ exceedance day), and increased VOCs (Figure 2c). Interestingly, the large increases in $NO_x$ during those evenings led to a substantial reduction in the VOCs-to-$NO_x$ ratio well below 5 that overlaps with the 25 ppb contour of $NO_x$ in Figure 2d, suggesting the relative importance of NO at the destruction of $O_3$ and the formation of $NO_2$. The set of reactions that pertain to the $NO_x$ cycle in R1-R3

$$NO + O_3 \rightarrow NO_2 + O_2 \tag{R1}$$

$$NO_2 + h\nu \rightarrow NO + O(^3P) \tag{R2}$$

$$O(^3P) + O_2 + M \rightarrow O_3 + M \tag{R3}$$

describe the role of NO as a sink to $O_3$ at night (R1) while photochemical reactions govern the dissociation of $NO_2$ (R2) and the production of $O_3$ as radical oxygen combines with molecular oxygen (R3). However, as discussed in the previous

paragraph, the reaction equations are incomplete since increases in $O_x$ and the VOCs-to-$NO_x$ ratio suggest a more complex set of chemical reactions with VOCs that lead to $O_3$ exceedance events that cannot be described by R2 alone, which includes the coupling of the $NO_x$ and $RO_x$ via R4-R7 rewritten from Wang et al. (2017)

$$HO_2 + NO \rightarrow OH + NO_2 \tag{R4}$$

$$RO_2 + NO \rightarrow RO + NO_2 \tag{R5}$$

$$OH + RH + O_2 \rightarrow RO_2 + H_2O \tag{R6}$$

$$RO + O_2 \rightarrow HO_2 + carbonyls \tag{R7}$$

Comparing the BL height in Figure 3e with $NO_x$ in Figure 2b during the evening shows relatively shallower BL heights when $NO_x$ was elevated compared to slightly deeper BL depths when $NO_x$ was low. Reductions in the nighttime BL are usually accompanied by light wind conditions, reduced wind shear, and cooler temperatures; however, it is challenging to make inferences about the representation of these meteorological quantities with BL height based on Figure 3. Nevertheless, a reduction in BL height, which is a result of increased static stability, can promote the removal of $O_3$ by $NO_x$ titration within a

reduced mixing volume under calm winds.

Figure 4 synthesizes results from Figures 2 and 3 by grouping data on days when $O_3$ exceeded 70 ppb (red) and during days when $O_3$ did not reach 70 ppb (blue). Much of what was discussed in Figures 2 and 3 is confirmed in Figure 4. However, there were also contrasts between Figure 4, and Figure 3 and Table 1 that are mentioned below.

Periods of high pressure (Figure 4i) coincided with increased temperature (Figure 4d), reduced relative humidity (Figure

4e), and a reduction in BL height (Figure 4g) and BL-averaged wind speed (Figure 4k) during the day, while at night, the surface temperature was cooler, BL heights were slightly shallower, and BL-averaged winds and wind speed shear (Figure 4h)

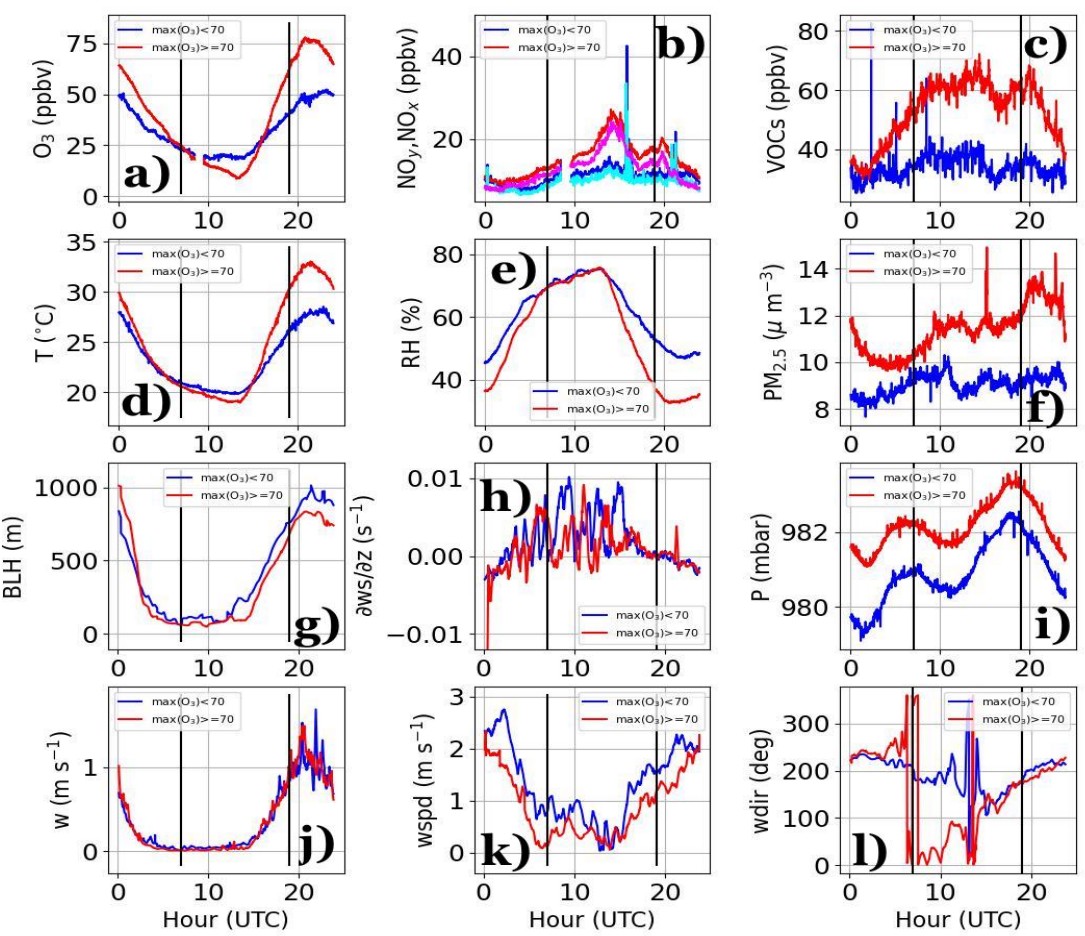

**Figure 4.** a) $O_3$, b) $NO_x$ (red and blue) and $NO_y$ (magenta and cyan), c) VOCs, d) surface temperature, e) surface relative humidity, f) $PM_{2.5}$, g) BL height, h) BL-averaged wind shear, i) surface pressure, j) BL-averaged vertical velocity, k) BL-averaged wind speed (wspd), and l) BL wind direction (wdir) derived by averaging the BL-averaged wind components $(u, v)$ grouped by days where $O_3$ exceeded (red or magenta–b)) and did not exceed (blue or cyan–b)) 70 ppb. Vertical black lines denote midnight (7 UTC) and noon (19 UTC).

was reduced. The reduction in wind speed and cooler temperatures at night when BL heights were lower supports increased static stability as mentioned in the previous paragraph. While little can be ascertained from the BL-averaged vertical velocity in Figure 4j, the wind direction in Figure 4l was northerly (southwesterly) during nights preceding days where $O_3 \geq 70$ ppb ($O_3 < 70$ ppb) before converging to a southwesterly wind into the afternoon hours in support of onshore flow.

The overall increase in surface pressure, which is known to promote fair weather conditions and periods of stagnation (Zhang et al., 2017), coincided with polluted conditions as shown by elevated $PM_{2.5}$ (Figure 4f), an increase in VOCs throughout the day (Figure 4c), and an increase in $NO_x$ (red) and $NO_y$ (magenta) during the night (Figure 4b) when averaging over $O_3$ exceedance versus non-exceedance days. The northerly winds observed during evenings preceding $O_3$ exceedance events (Figure 4l) may have contributed to increased biogenic VOCs (Figure 4c) advected from the San Gabriel Mountains and increased $PM_{2.5}$ from lingering wildfire smoke. Averaging over $O_3$ exceedance periods when $PM_{2.5}$ and VOCs was generally high also occurred contemporaneously with lower BL heights, weaker wind conditions, and increased temperature during the day according to results in Figure 4. Though increased surface temperature is usually accompanied by deeper BL heights, increased surface pressure typically favors large-scale subsidence that could lead to warming aloft, thereby increasing static stability at the height of the BL layer inversion. The relative strength of the superadiabatic layer near the surface and the strength of the BL inversion act as controls on the BL height through positive and negative buoyancy, respectively. While it is clear that the BL structure is different between low ($O_3 < 70$ ppb) and high ($O_3 \geq 70$ ppb) pressure days, it must also be kept in mind that an increase in temperature could lead to higher use of cooling facilities that can promote an increase in pollution (Zhang et al., 2017).

Other notable features in Figure 4 are found in the wind speed shear (Figure 4h), spikes in $NO_x$ and $NO_y$ (Figure 4b), and the semi-diurnal pattern in surface pressure. The increased wind speed shear at night across the BL is a mechanism for enhanced mechanical production of turbulence that can deepen the BL and weaken stability, thus supporting the concomitant increase in BL heights in the evening as previously mentioned (Figure 4g). The wind speed shear during the day was generally negative as a result of a developing near-surface onshore flow hosting a wind maximum well within the convective BL. The spikes in $NO_x$ occurred some time after sunrise (16 UTC) and around the time period SBs arrived (21 UTC or 2p PT). However, these spikes could represent exceptional events that are not representative of all days and yet stand out due to a relatively small sample size. The semi-diurnal pressure pattern occurred throughout the whole month of August. Local peaks in pressure occurred at 7 UTC (midnight PT) and 19 UTC (noon PT), respectively, with a general increase in pressure that began around sunset (around 0 UTC). Other panels in Figure 4 do not feature this semi-diurnal trend; however, the troughs in pressure at 1 UTC (6p PT) and 11 UTC (4a PT) occurred near the evening and morning transitions, respectively.

The reduction in BL height and an increase in surface pressure noted earlier is interesting given that these quantities were found to be uncorrelated (Table 1). Reviewing Figure 3, it is clear that BL heights tended to be more elevated during non-exceedance days, while at least half the days during $O_3$ exceedance events overlapped with low BL heights. Similarly, $O_3$ exceedance days overlapped more with moderate-to-high pressure conditions, while non-exceedance days overlapped with a mix of low and high pressure days. Other notable departures between Figure 4 and Figure 3 relate to the occasional pattern mismatch between pressure, $PM_{2.5}$, and VOCs. In Figure 4, all three quantities were elevated during $O_3$ exceedance days, while in Figure 3 there were days where $O_3$ exceeded 70 ppb and yet did not coincide with increases in these quantities. Thus, while averaging across $O_3$ exceedance versus non-exceedance days may bring out different information related to statistics, it is also important to note that the averages may be skewed to extremes associated with a temporally limited data set. Furthermore, each day could offer a unique set of chemical and meteorological conditions that lead to challenges when making generalizations.

## 4 Evaluating the Micrometeorological Role on Air Quality, a Case Study – 16 August 2021: Pasadena, CA

In the previous section we evaluated the diurnal structure of $O_3$, $NO_x$, and VOCs during August 2021 and their relationship to the large-scale meteorology and BL structure. We now turn to a case study highlighting the convective growth phase superimposed with a SB that occurred on 16 August (i.e., gold dashed line in Figures 2 and 3) to study the micrometeorological impacts on air quality measurements. While a SB was observed most days during the month of August – winds transitioned to a southerly-to-southwesterly regime with BL wind enhancement (refer to Figure 3c and Figure 4k for wind speed increases and Figure 4l for wind direction shift around noon, 19 UTC) – we chose this case study to examine the temporal variability observed in $NO_x$ and $O_3$ that was 180 degrees out of phase following the arrival of the SB (discussed below), and to evaluate the utility of the MSCM under different BL transitions that occurred over a single diurnal period.

### 4.1 Synoptic and Regional Conditions

The LA basin was largely free of major large-scale meteorological features. The 500 mb upper air map (https://weather.uwyo.edu/cgi-bin/uamap?REGION=naconf&OUTPUT=gif&TYPE=obs&TYPE=an&LEVEL=500&date=2021-08-16&hour=12) shows a tropical cyclone far to the south and a semi-permanent offshore high pressure system over the Pacific ocean, east of Hawaii. The extratropical patterns farther to the north did not extend to the LA basin. The surface analysis map (https://www.wpc.ncep.noaa.gov/archives/web_pages/sfc/sfc_archive_maps.php?arcdate=08/16/2021&selmap=2021081612&maptype=namussfc) reveals a more complex dynamical set-up. Over the southwest portion of the United States was a mix of local high and low pressure systems that exhibited a wave-like character behind a dryline. Two weak low pressure centers were in relatively close proximity to the LA basin at this time, with the low east of the LA basin adjacent to the alternating low and high pressure centers positioned behind the dryline and an inverted trough over central California to the north.

Figure 5 shows the HRRR output of BL height reported above ground level (shading), near-surface winds (white barbs), and 500 mb Geopotential height (gray contours). Overlaid on maps are observations of $O_3$ (shaded circles) from AIRNOW and the Pasadena site location as indicated by a magenta star. BL heights in the morning were generally low (sub-kilometer) across the LA basin and across large swaths of elevated terrain, though local increases in BL height in excess of 2 km were evident at 17 UTC (10a PT) (Figure 5b). The winds over the LA basin during the morning hours were weak, with little in the way of a discernible wind pattern. $O_3$ concentrations were generally low across the LA basin with some increases to the north and east of Pasadena over elevated terrain. Winds over the coastal ocean were nearly uniform and followed a cyclonic pattern along the coastline.

By the afternoon, $O_3$ began to increase near the coastline and across the LA basin. BL heights also increased in the LA basin, with some areas locally increasing in BL depth to about 1.2 km. Farther north into elevated terrain, BL heights increased from 1.5 km at 12p PT to 2.5 km at 2p PT. A southwesterly flow that developed in the early afternoon penetrated farther into the LA basin towards elevated terrain by 2p PT, with a clear drop in BL height across coastal areas as marine BL air propagated into the region. Enhancements in $O_3$ between 19 and 21 UTC (12p and 2p PT) occurred east of Pasadena, with concentrations in excess of 100 ppb. Observations of onshore flow reported by the DL occurred at 18 UTC (11a) as winds transitioned from

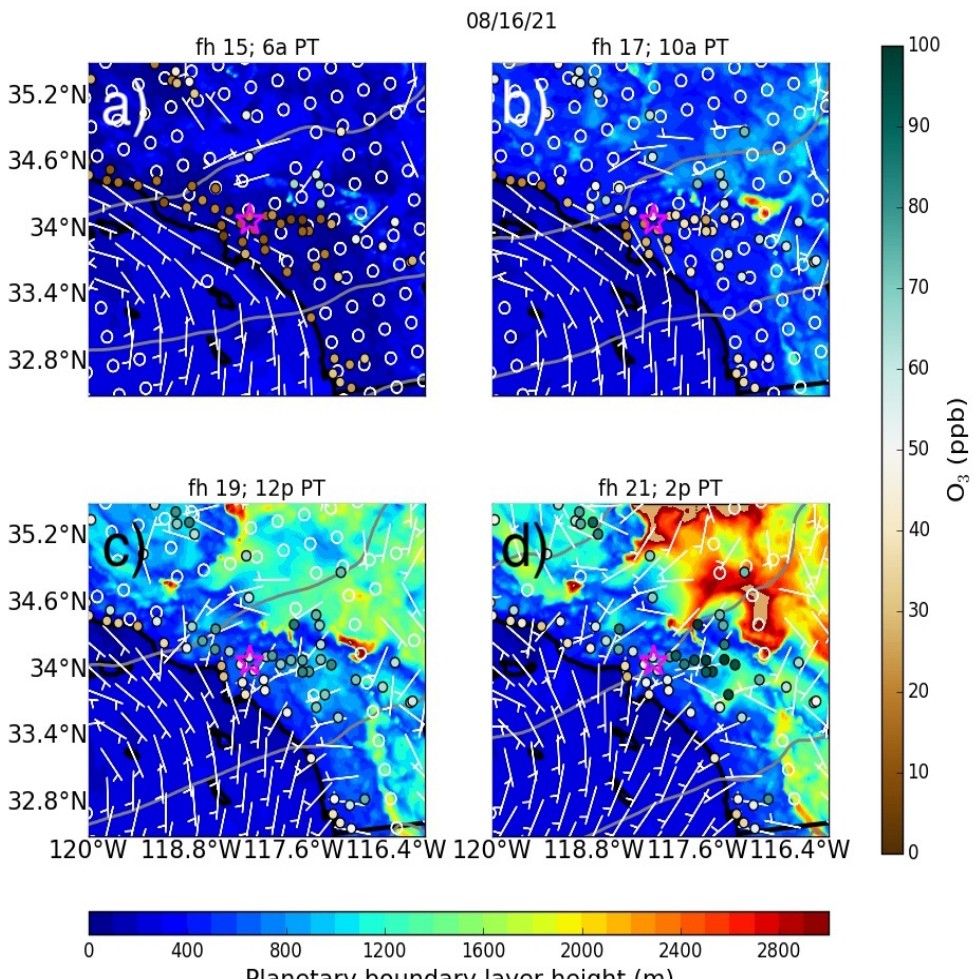

**Figure 5.** HRRR output of BL height above ground level (shading), near-surface winds (white wind barbs), 500 mb Geopotential height (gray contours), AIRNOW $O_3$ observations (circles–shaded), and the Pasadena site location (magenta star) during a) 15 UTC (6a PT), b) 17 UTC (10a PT), c) 19 UTC (12p PT), and d) 21 UTC (2p PT).

an easterly to southerly wind along with increased wind speed within the BL (Figure 7a and orange square overlaid on BL-averaged wind speed in Figure 7d) that was in relatively close agreement with timing of onshore flow predicted by the HRRR. The enhancement in $O_3$ east of Pasadena occurred along the leading edge of southwesterlies and a gradient in BL height as a result of onshore flow associated with a SB, which can form a convergence line. The enhancement of $O_3$ from AIRNOW along

the leading edge of southwesterlies by the HRRR corroborates findings from Nauth et al., which also found poor air quality conditions within the vicinity of a convergence line on SB days. However, it is important to note that the BL height gradient occurred across elevated terrain, where the backdrop of the mountains can act as a natural barrier to SB penetration, thereby leading to elevated pollution levels in situations where flow propagation becomes limited by the potential energy associated with ascending elevated terrain, and because of flow interactions with neighboring pressure patterns as discussed earlier.

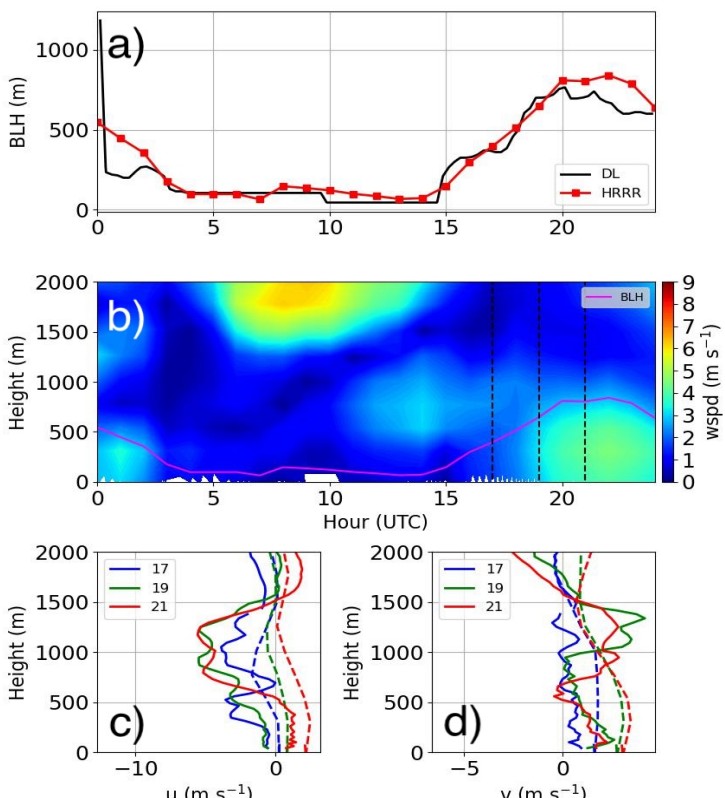

**Figure 6.** a) A comparison between BL height reported by the DL (black) and the HRRR (red), b) a time-height cross-section of wind speed from the HRRR with BL height (magenta) and selected times to analyze profiles in c) and d) (black vertical dashed lines), and a comparison between the HRRR (dashed) and the DL (solid) for the c) zonal and d) meridional wind encompassing the SB transition during 16 August, 2021.

### 4.1.1 DL versus the HRRR

To support using the day 1 forecast from the HRRR to examine the BL wind structure, we compared BL height and the component wind profiles at selected times encompassing the SB transition (dashed lines in Figure 6b). Figure 6a gives confidence in relying on the HRRR to model the diurnal structure of BL height despite slightly overpredicting BL heights following the SB transition by about 100 m. The zonal wind from the HRRR in Figure 6c (dashed profiles) followed the observed structure (solid profiles) reasonably well, but failed to capture the details of vertical variations in the wind, and generally favored a westerly wind over observed easterly winds spanning the period encompassing the SB transition (dashed lines in Figure 6b). The meridional wind in Figure 6d captured enhanced southerly flows that were distributed much more broadly in the HRRR (dashed profiles) than observations from the DL (solid profiles), with the latter revealing the development of a shallow and weak low-level jet (LLJ). Above the BL, the HRRR failed to capture wind enhancements observed by the DL, which was is also evident when comparing wind speed in Figure 6b with Figure 7a. While winds above the BL were inadequately captured by the HRRR, the BL height, the timing of onshore flow, and the enhancement in southerly winds indicated that HRRR correctly predicted the arrival of the SB between 18 and 19 UTC (11a and 12p PT) and the overall BL dynamics. Other differences between the HRRR and the DL were found in the near-surface wind direction, which in the case of the HRRR exhibited a consistently southwesterly flow while the DL reported a transition from south-southeasterlies to a southwesterly wind.

### 4.2 In Situ Chemistry and Doppler Lidar Observations

Figure 7 shows the horizontal wind speed and wind direction overlaid with a 1 m s$^{-1}$ vertical velocity contour and BL height from the DL (a-b); in situ $NO_x$, VOCs, $O_3$, and $O_x$ (c); and BL-averaged wind speed and in situ temperature, dew point, and pressure (d) in Pasadena, CA. A shallow BL developed during the evening (less than 60 m) with weak easterly winds within the first 500 m that were occasionally interspersed with shallow northerly flows extending across the first 100 m from the surface. Above 500 m, winds increased and veered northwesterly, likely as a result of combined influences of clockwise flow associated with the offshore high pressure system and counter-clockwise flow from the inverted trough converging over the San Gabriel Mountains to the north. VOCs and $NO_x$ increased into the night as the BL became shallower. The decrease in BL height also coincided with reduced temperatures and a drop in $O_3$. The inflected behavior between $NO_x$ and $O_3$ throughout the evening is revealed by nearly constant $O_x$ as shown in gray.

The BL morning transition began a little after 13 UTC (6a PT) as the temperature increased, the BL deepened, and winds near the surface transitioned from northerly to southeasterly. It was at this time that $O_3$, $NO_x$, and VOC concentrations changed rapidly. The initial response in VOCs and $NO_x$ was similar: a brief increase followed by a sharp decrease. $O_3$, by contrast, increased over the same time interval that $NO_x$ and VOC concentrations decreased. Between 15 and 18 UTC (8a and 11a PT), $O_3$ exhibited a linear increasing trend that followed closely with surface temperature. A second decrease in VOC concentrations around 17 UTC (10a PT) coincided with a sharp drop in $NO_x$ and a slight lull in BL growth, and occurred simultaneously with a change in wind direction from southeasterly to southerly. At the same time, an easterly wind appeared to descend from above (gray arrows in Figure 7a-b) along with increased horizontal winds at BL top (1 km). We hypothesize that de-

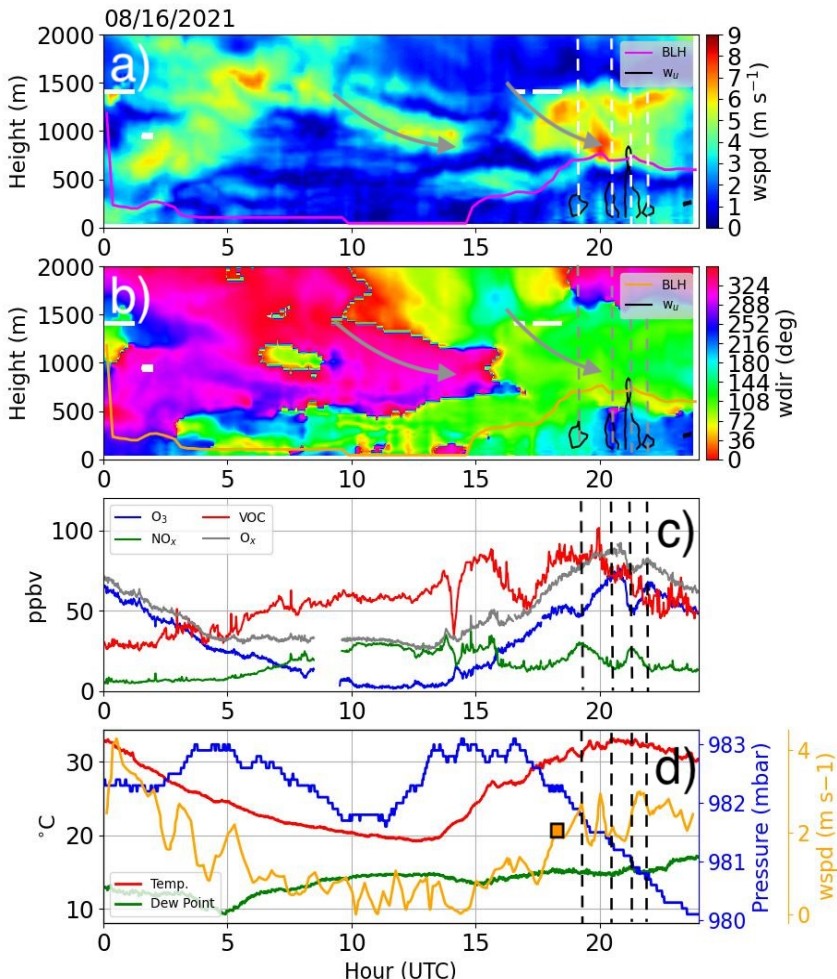

**Figure 7.** Profiles of a) wind speed and b) wind direction, c) in situ $O_3$ (blue), $NO_x$ (green), VOCs (red), and $O_x$ (gray), and d) BL-averaged wind speed (orange), temperature (red), dew point (green), and pressure (blue) in Pasadena during 08/16/21. Overlaid in a) and b) are gray arrows indicating descent (i.e., subsidence). Contours of 1 m s$^{-1}$ of vertical velocity are included in a)-b) along with BL height. Dashed lines intersecting the center of updrafts in a) white, b) gray, and c-d) black are also shown. An orange square is overlaid in d) to highlight a dynamical transition featuring wind enhancement and a rotation from an easterly to southerly flow that occurred around 18 UTC (11a PT). The consistent upward vertical motion in c) is suspected to be related to the smearing of weak downward motions relative to stronger upward motions over 15-minute time intervals.

scending winds from aloft led to an increase in the strength of the BL inversion as a result of adiabatic compression initiated by subsidence, which can limit entrainment/detrainment between the BL and free troposphere. Therefore, it is speculated that the enhancement in winds above the inversion resulted from this mechanism, and that a conversion of vertical momentum to horizontal momentum occurred as descending wind jets encountered the inversion. Regional soundings at Vandenberg Air Force Base (https://weather.uwyo.edu/cgi-bin/sounding?region=naconf&TYPE=GIF%3ASKEWT&YEAR=2021&MONTH=08&FROM=1600&TO=1612&STNM=72393) and San Diego (https://weather.uwyo.edu/cgi-bin/sounding?region=naconf&TYPE=GIF%3ASKEWT&YEAR=2021&MONTH=08&FROM=1600&TO=1612&STNM=72293) near these times reveal a large dew point depression above the near-surface thermodynamic inversion that is indicative of subsidence. A more detailed analysis of descending wind jets is reserved for Appendix C.

Changes in the wind structure at the top of the BL covaried with small fluctuations in BL height, BL winds (horizontal and vertical), and, to some extent, pollution concentrations (refer to dashed lines in Figures 7). The increase in updraft strength shown by the contours in Figure 7a-b coincided not only with increased winds at BL top and a transition to stronger surface winds (orange square overlaid on BL-average wind speed in Figure 7d) with a southerly component around 18 UTC (i.e., arrival of the SB), but also bursts in wind speed that were sometimes staggered temporally with updrafts. The spacing between updrafts occurred coincidently with temporal extrema in $NO_x$ and $O_3$, and is believed to be related to the transport dynamics associated with the SB propagating into Pasadena as well as the development of strong wind shear across the BL that likely promoted additional entrainment at the top of the BL.

The pulsing/periodic development of updrafts appear largely responsible for the 'oscillations' in $NO_x$ and $O_3$, which were found to be 180 degrees out of phase. Examining $O_x$ during this time reveals small amplitudinal variations sharing the same phase as $O_3$ (i.e., $O_3$ fluctuates with greater amplitude than $NO_x$). Unlike $NO_x$ and $O_3$, VOC concentrations did not feature these oscillations, and in fact gradually decreased as the surface temperature and dew point decreased and increased, respectively. The decrease (increase) in surface temperature (dew point) is likely related to cooler and moister marine BL air entering Pasadena. A decrease in VOCs could be related to the introduction of an air mass with less VOCs, or as a result of interactions with unrepresented chemistry not considered in this analysis. However, as previously mentioned, the sinusoidal-like variations that began at the onset of the SB are visually correlated with the micrometeorological characteristics of the BL noted by instances of increased vertical motion and wind speed bursts in Figure 7. Despite variations in $NO_x$ and $O_3$ appearing correlated with the SB dynamics, we cannot rule out other factors such as chemical reactions within an altered mixing volume and differential advection.

### 4.3 Spectral Characteristics of Air Quality and Meteorological Variables

Some of the fine structure details observed in in situ chemistry measurements visually correlated with micrometeorological features associated with evolving BL conditions, especially during the growth phase of the BL and during the onset of the SB. Using techniques to remove contributions from the diel cycle, we now focus more quantitatively on the micrometeorological influence on pollution concentrations using methods described in sections 2.2.1-2.2.4.

Figure 8 shows the temporal variability of chemistry ($O_3$, $NO_x$, VOCs, and $O_x$–a-c, f) and meteorological (temperature, relative humidity, BL-averaged wind speed, BL-averaged wind shear, and BL height – d-e, g-i) measurements as reported in scaleograms. Peaks in $O_3$ related to the fine-scale structure were first observed at 19 UTC (noon) and are related to the sinusoidal-like variations observed in Figure 7c during the onset of the SB as the height of the BL height climaxed. The frequency of the fine structure related to sinusoidal-like variations in $O_3$ increased, which led to a decrease in the temporal width of variations from a half hour to 15 minutes. Similar peaks in $NO_x$ (Figure 8b) and $O_x$ (Figure 8f) scaleograms were also observed during this time, but with $O_x$ departing in the time evolution of extrema relative to $O_3$ and $NO_x$ by exhibiting nearly constant temporal variations that were on the order of 20 minutes. Other variables did not appear strongly correlated with fine structure characteristics of $O_3$ based on visual inspection.

The decrease in temporal width of the periodic-like structure in $O_3$ and $NO_x$ is interesting, and if we examine the BL-averaged wind direction during the time that these oscillations were observed in Figure 7d, it becomes clear that the decrease in temporal width of extrema coincided with an increase in BL-averaged wind speed. It is suspected that the decrease in temporal width is related to increased winds with respect to time. An analysis dedicated to exploring this relationship is left as an exercise in Appendix D.

The morning transition featured robust variability in air quality measurements, particularly with respect to $NO_x$ and VOCs (Figure 8b-c). Small-scale variability near sunrise was observed in relative humidity (Figure 8e), BL-averaged wind speed (Figure 8g), BL height (Figure 8i), and $NO_x$ (Figure 8b) and VOCs (Figure 8c) as surface heating promoted BL growth and the erosion of the residual layer through entrainment. The micrometeorological response near the surface manifested as high frequency peaks between 5 and 30 minutes, with $NO_x$ and VOCs exhibiting changes in concentration at a higher frequency than dynamic and thermodynamic measurements. Undoubtedly, the rapid mixing and entraining of the residual layer that initially began across a shallow BL not only led to adjustments in the wind speed and the thermodynamic vertical structure, but also the depth of the BL and the pollutant distribution extending across the BL as air from the residual layer was entrained during the BL growth phase. The temporal width of $NO_x$ and VOCs varied less rapidly as the BL deepened, with time-scales ranging from 10 minutes at sunrise to a little over an hour as the BL reached maximum depth that is somewhat matched by BL height variations shown in Figure 8i.

Other features in Figure 8 include small-scale variability in temperature near the time that the SB arrived to Pasadena; the nighttime variability in relative humidity, temperature, and wind shear during the evening transition; and larger temporal variations observed in relative humidity (Figure 8e) that increased into the evening before the onset of rapid mixing at sunrise that led to rapid changes in surface meteorology measurements.

Using the algorithmic approach outlined in Section 2.2.4 and Appendix A, we now seek variables that share spectral charac-teristics with $O_3$, $NO_x$, VOCs, and $O_x$. We avoid pairing in situations where there were strong self-correlations (e.g., $O_x$&$O_3$ and $O_x$&$NO_x$) and determine pairings with the strongest spectral similarities. Figure 9 shows the spectral structure of different variables either paired with $O_3$, $NO_x$, or VOCs; $O_x$ did not make the list of the top 9 variable pairings. With the exception of $O_3$&$NO_x$, which was added to the plot independently, the remaining pairings are in order of structural similarity or maximum pattern likeness from b) to i). The spectral structure of $O_3$&$NO_x$ in Figure 9a highlights the covariability between $O_3$ (Figure

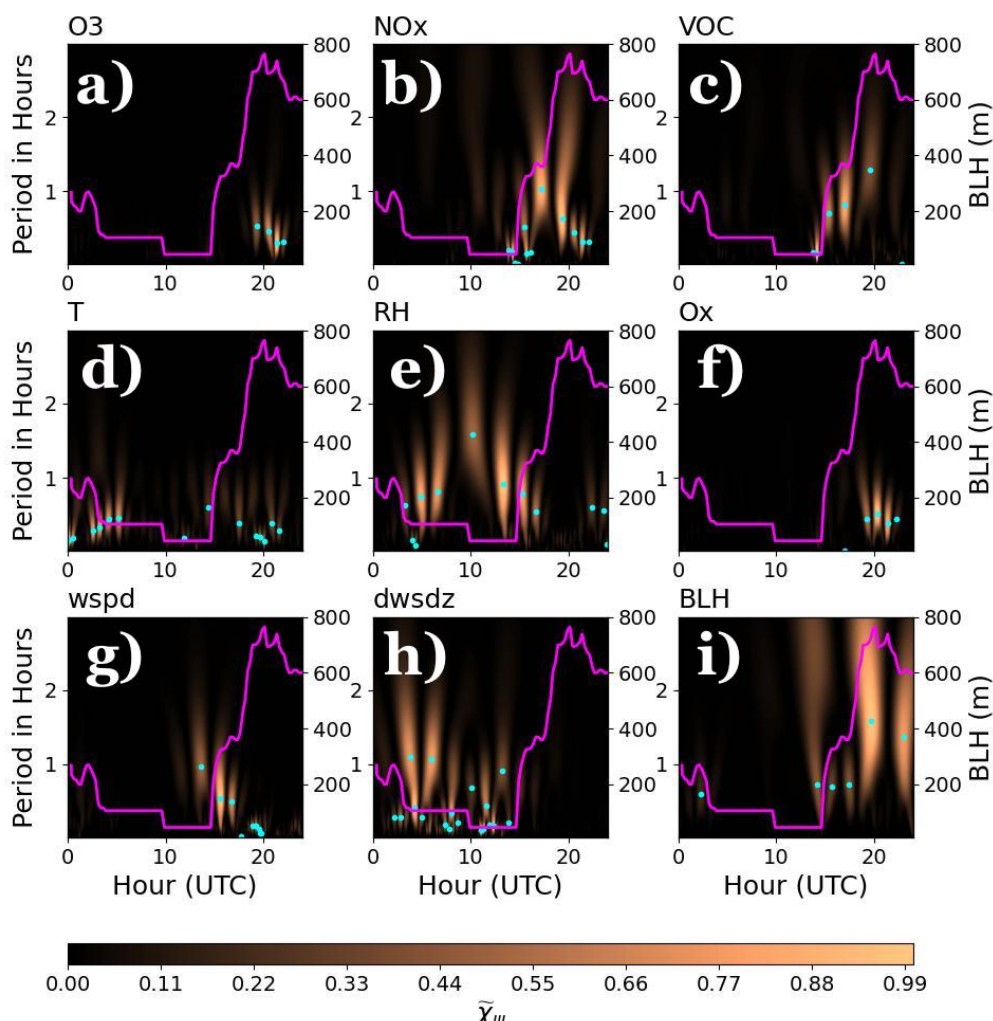

**Figure 8.** Scaleograms of a) $O_3$, b) $NO_x$, c) VOCs, d) temperature (T), e) relative humidity (RH), f) $O_x$, g) BL-averaged wind speed (wspd), h) BL-averaged wind speed shear, and i) BL height during 16 August. Overlaid on scaleograms is BL height in magenta and the location of maxima in the power spectral peaks shown by cyan dots.

8a) and $NO_x$ (Figure 8b) during the SB transition, where only the peaks sharing similar temporal widths coincident in time

stand out between the two variables. The overlapping spectral structure featuring dominant peaks (well above 0.8) is related to the sinusoidal-like variations in $O_3$ and $NO_x$ discussed in Figure 7c that coincided with the spacing of updrafts (Figure 7a-b). Other chemistry pairings revealed similarities between VOCs with $NO_x$ during the growth phase of the BL. For instance, the maximum normalized PSD in Figure 9d featured less rapid temporal variations in the fine structure as the BL deepened, thus hinting at the role of increased overturning time-scales associated with mixing across a deeper layer. The meteorological vari-

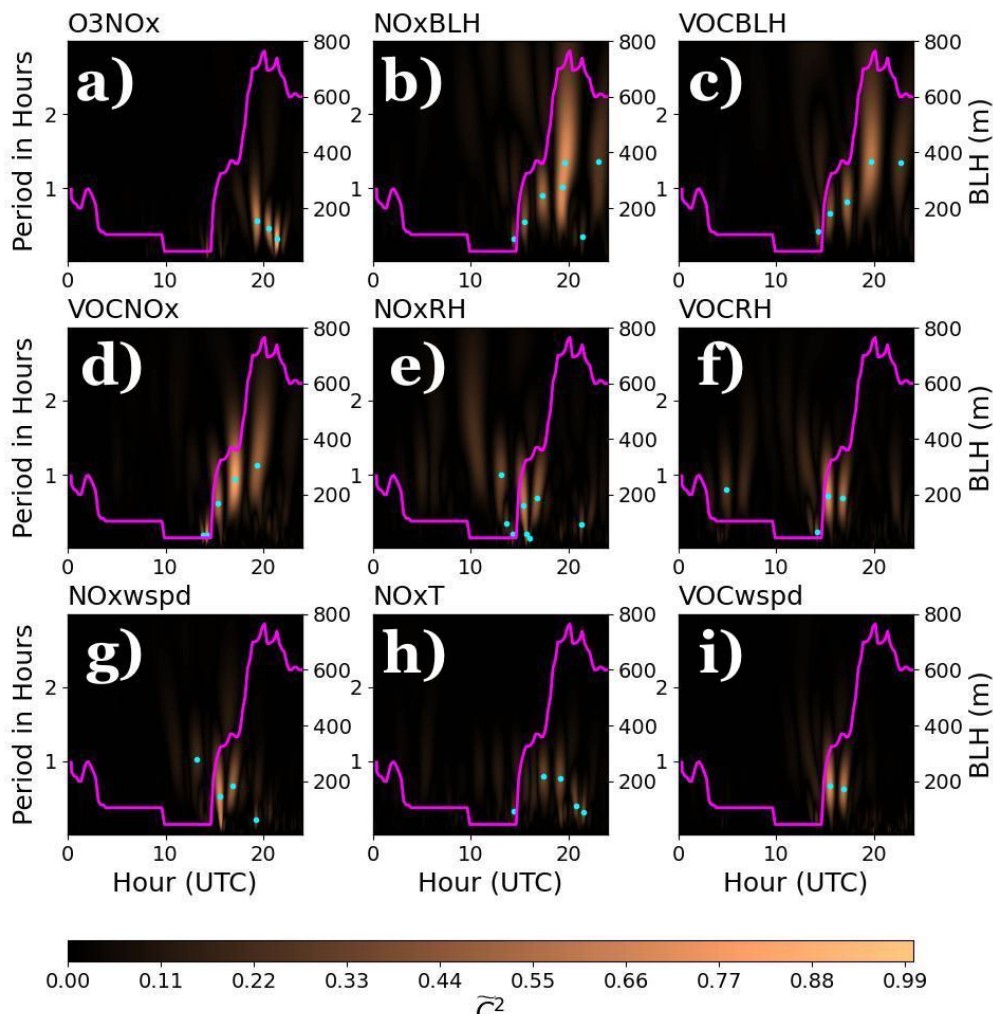

**Figure 9.** Scaleograms of a) $O_3$&$NO_x$, b) $NO_x$&BL height, c) VOCs&BL height, d) VOCs&$NO_x$, e) $NO_x$&relative humidity, f) VOCs&relative humidity, g) $NO_x$&BL-averaged winds speed, h) $NO_x$&temperature, and i) VOCs&BL-averaged wind speed on 08/16/21. Overlaid on scaleograms is BL height in magenta and the location of maxima in the power spectral peaks shown by cyan dots.

able whose spectral structure agreed the most with in situ chemistry observations was BL height (b-c). In fact, the cross-spectra between VOCs&BL and $NO_x$&BL is remarkably similar, with the time-scale of variability ranging from 15 minutes during sunrise to about 1.25 hours as the BL climaxed. The similarity between VOCs&BL and $NO_x$&BL should not be too surprising, however, given the spectral structure in Figure 9d.

The fine structure of other meteorological variables, such as relative humidity, temperature, and BL-averaged wind speed,
matched well with the fine structure in $NO_x$ and VOCs during sunrise. The initial growth of the BL naturally led to an adjust-

ment in the relative humidity and wind structure as drier air and stronger winds from aloft mixed to the surface. Covariations during the initial growth phase between $NO_x$ and VOCs with relative humidity and BL-averaged wind speed (Figure 9e-g, i) occurred over 5 to 45-minute periods between 15 and 17 UTC (8a and 10a PT), with pairings including wind speed leading to larger temporal widths (DL had coarser resolution). Faint spectral signatures are also evident in $NO_x$&temperature measure-

ments that line up with temporal variations in relative humidity&BL-averaged wind speed during the initial BL growth phase and during the onset of the SB (after 20 UTC or 1p PT). The faint spectral signatures, however, must be taken lightly since a reduced PSD between two variables implies reduced coherence between the temporal structure of variables paired.

Combinations of three variables were also examined as shown in Figure 10. While interdependencies are reduced when grouping multiple variables, VOCs&$NO_x$&BL height variations in Figure 11a showed a strong dependence between these

three variables during the BL growth phase according to the maximum normalized PSD, which is in agreement with findings in Figure 10b-d. Other combinations between three variables reveal rapid changes during the initial growth of the BL.

The spectral characteristics from Figures 8-10 highlight the role of BL growth and a SB transition on modifications to chemistry and dynamics measurements, and how the time-scale of air quality measurements changes as the BL evolves. The clearest impacts occurred during the BL growth phase. As such, we isolate variables and pairings of variables in Figures 8 and

9, respectively, that showed a decrease in time-scale as the BL height increased. Figure 11a plots cyan dots from Figures 8 and 9 (i.e., the temporal extremum) for variables and variable pairings that were sensitive to BL growth, color-coded according to the legend, and fitted using an assumed powerlaw. With the exception of $NO_x$, all other variables and variable pairings were in close agreement with one another. Although $NO_x$ is an outlier, the trend for each variable and variable pairing shows an increase in temporal extremum from about 0.2 hours to a little over an hour spanning the BL growth phase. We now combine

the BL height reported at the time that a temporal extremum within scaleograms was identified during the BL growth phase, i.e.,

$$V_{\tau_{max}} = \frac{z_{BL}}{\tau_{max}} \tag{12}$$

where $V_{\tau_{max}}$ is the BL height growth rate. Figure 11b combines the numerator (y-axis) and denominator (x-axis) to derive the slope ($V_{\tau_{max}}$). As can be seen, the reported slopes range between 0.14 m s$^{-1}$ to 0.21 m s$^{-1}$, with the weakest confidence in

$NO_x$–all other variables correlate well with the linear fit (i.e., >0.9). The estimated growth rate is quite a bit higher compared to the slope of 0.03 m s$^{-1}$ derived by applying a linear fit to BL height in Figure 11c. However, an examination of Figure 12 reveals that $NO_x$ and VOC extrema alternated between troughs and peaks during the BL growth phase, while at the same time exhibiting a reduction in amplitude and a broadening in temporal variability. The temporal spacing observed peak-to-peak (and trough-to-trough) ranged between 2.5 and 5 hours (e.g., can be seen by eyeballing Figure 12d), while $\tau_{max}$ ranged between 0.1

and 1.2 hours, which, if the extremes defining these respective ranges are averaged and divided (i.e., $\overline{\tau}/\overline{\tau}_{max}$), yields a factor close to $2\pi$. Acknowledging that the growth phase of the BL is a combination of developing convection and mixing down of the residual layer air aloft, and that eddies within the BL inherently have vortical motion and therefore rotate air, we use this argument and results from Figure 12 to normalize the slopes reported in Figure 11b by $2\pi$. Adopting the $2\pi$ normalization factor and noting constant velocities derived from the slopes leads to agreement with d$z_{BL}$/dt during BL growth phase such

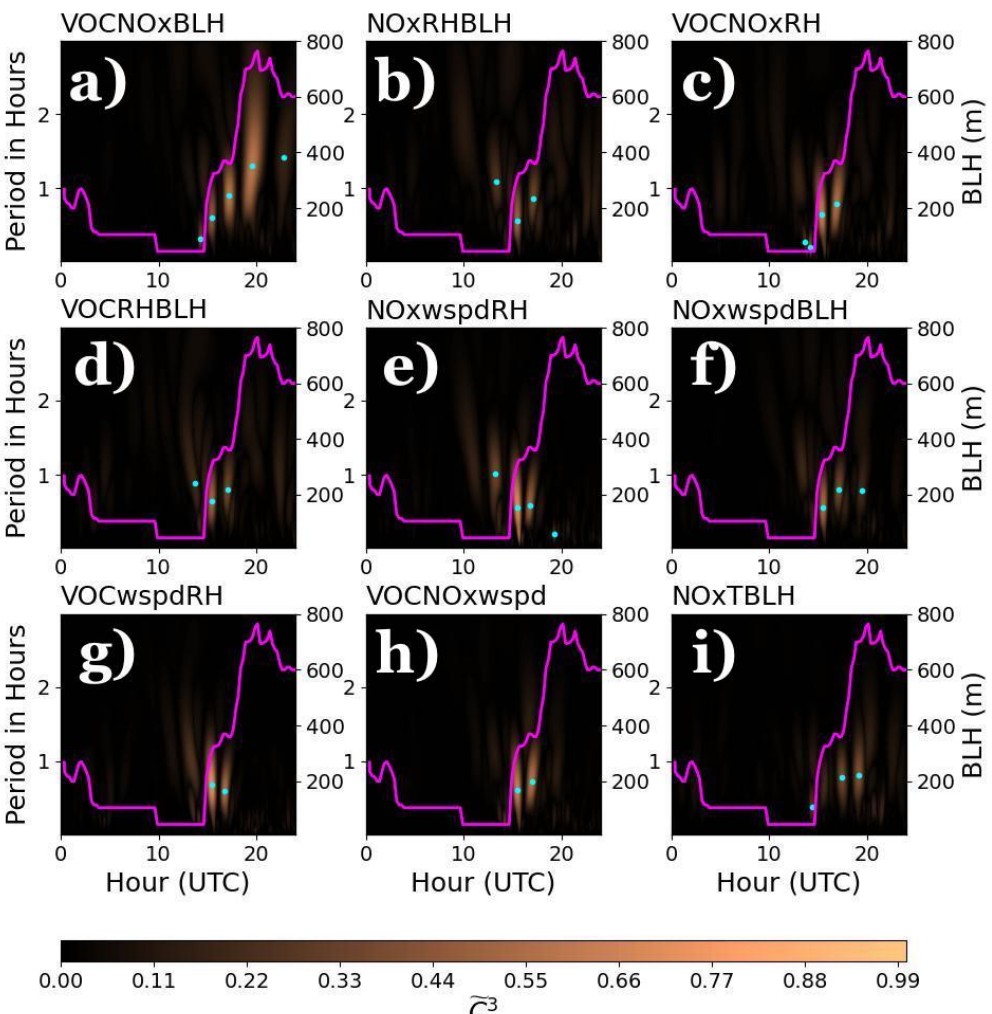

**Figure 10.** Scaleograms of a) VOCs&$NO_x$&BL height, b) $NO_x$&relative humidity&BL height, c) VOCs&$NO_x$&relative humidity, d) VOCs&relative humidity&BL height, e) $NO_x$&BL-averaged winds speed (wspd)&relative humidity, f) $NO_x$&BL-averaged wind speed&BL height, g) VOCs&BL-averaged wind speed&relative humidity, h) VOCs&$NO_x$&BL-averaged windspeed, and i) $NO_x$&temperature&BL height on 08/16/21. Overlaid on scaleograms is BL height in magenta and the location of maxima in the power spectral peaks shown by cyan dots.

that

$$\frac{dz_{BL}}{dt} \approx \frac{z_{BL}}{2\pi\tau_{max}} \tag{13}$$

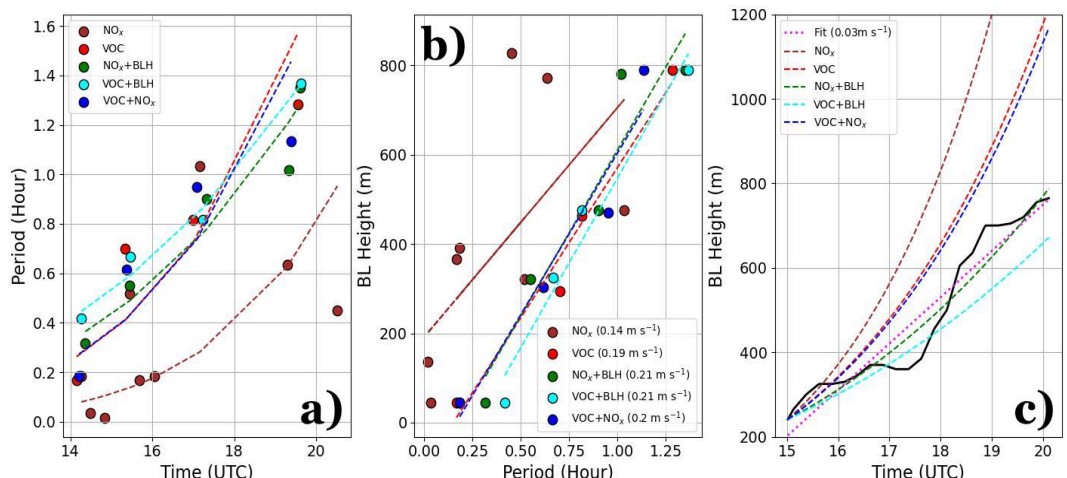

**Figure 11.** a) Time versus maximum period ($\tau_{max}$), b) maximum period versus BL height, and c) time versus BL height. In a), b), and c) we overlay lines of best fit using an assumed powerlaw, linear fits, and fits using Eq. (15), respectively. A pink dotted line in c) is additionally included as a linear fit to BL height with a derived growth rate of 0.03 m s$^{-1}$.

Eq. (13) approximately holds for the case analyzed and represents a first order differential equation, the solution of which is exponential under the assumption that $\tau_{max}$ does not change with respect to time, i.e.,

$$z_{BL}(t) = z_{BL,0} \, e^{t/2\pi\tau_{max}} \tag{14}$$

However, because $\tau_{max}$ changes with time, then Eq. (14) cannot be used as a valid analytical function to model the behavior of the BL. Instead, if we revisit Eq. (12), take the time derivative, carry out a series of algebraic manipulations and substitutions, integrate with respect to time, and assume that the estimated BL height growth rate does not change appreciably in time as justified by the lines of best fit in Figure 11b, then we arrive to Eq. (15)

$$z_{BL}(t) = z_{BL,0} \frac{\tau_{max}(t)}{\tau_{max,0}} \tag{15}$$

where $z_{BL,0}$ is the BL height at sunrise and $\tau_{max,0}$ is the time-scale at sunrise derived from fits in Figure 11a. A full derivation of Eq. (15) is left for Appendix B.

As can be seen in Figure 11c, the fits have a wide range of behavior, with several overestimating, one slightly underestimating, and another modeling the BL growth with strong confidence. The strong sensitivity between modeled curves is owed to the different estimates of $\tau_{max}$ between different variables and variable pairings examined and the derived power from the fits

in Figure 11a. The modeled curves that closely reproduce the evolution of the BL are from $\tau_{max}$'s derived when combining the 2D spectral structure from chemistry output with BL height variations rather than single variables alone (i.e., NO$_x$ and VOCs). While this appears borne out of hyper-dependence on BL height, we are simply using the fact that the temporal structure of NO$_x$ and VOCs during the growth phase is highly correlated with variations in BL height rather than BL height itself. The overestimating fit lines closest to the observed BL height trend (VOCs&NO$_x$ and VOCs) model the BL height reasonably well

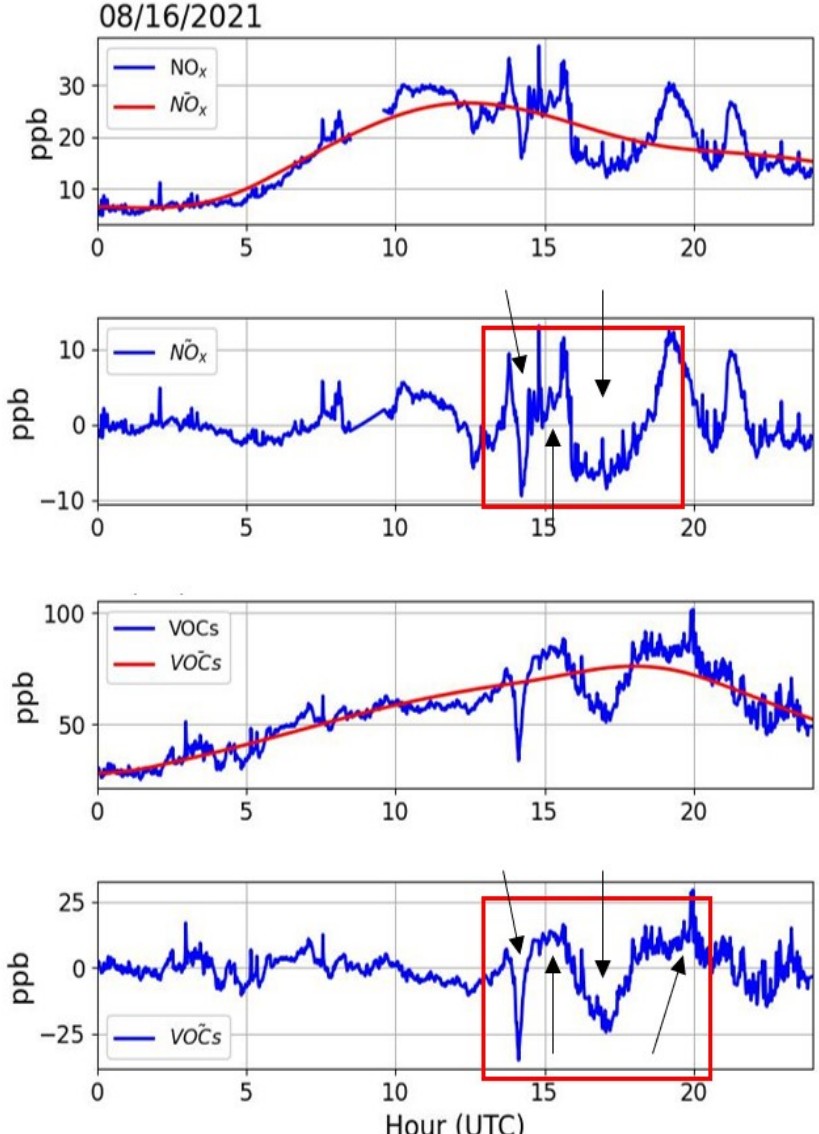

**Figure 12.** a) In situ measurements of NO$_x$ overlaid with the trend of NO$_x$ with fine-scale variations removed, b) the fine-scale variations of NO$_x$, c) in situ measurements of VOCs overlaid with the trend of VOCs with fine-scale variations removed, and d) the fine-scale variations of VOCs. The red boxes and arrows in b) and d) highlight peaks and troughs during the BL growth phase that reduce in amplitude and broaden in the temporal structure.

up to 18.5 UTC (11:30a PT) before departures between fits and the observed BL become large, which occurs approximately at the start of the SB transition. Adjustments not only in the mixing volume and the depth of the BL, but changes in chemical reactions as concentrations dilute during BL growth and mix free tropospheric air from above offers a possible explanation

for the mismatch between observed and modeled BL height. Other possible factors include advection as the wind direction changed. However, winds were generally weak in the BL with gradual veering during the BL growth phase.

The results above highlight variables affected during BL growth and during the onset of the SB. While $NO_x$ and VOCs showed a response that scaled with the BL, $NO_x$ and $O_3$ were sensitive to the dynamical evolution of the SB. Mapping the spectral structure of variables onto one another allowed an examination of shared temporal characteristics that further accentuated the role of BL on the fine structure details of pollutants during transitional periods, and allowed quantitative estimates of the temporal variability of measurements. However, one must wonder the representativeness of this finding? For that, we now turn to Section 5 to explore this problem more statistically.

## 5   BL Growth and Transitions During August 2021

Applying the methods described in 2.2.1-2.2.4 to the entire dataset allows a statistical evaluation of the BL growth phase for the month of August. We isolate the time period spanning 14 and 20 UTC (7-13 PT), identify all the dominant temporal extremum from scaleograms as discussed in Section 2.2.3, and determine how the temporal width of extrema varied with time and BL height. We consider variables and variable pairings that exhibited rapid changes during BL growth in Figures 8 and 9.

Figure 13 represents the temporal width of extremum (x-axis) versus BL height (y-axis) for selected variables and variable pairings. The size of the scatter indicates time of day during the BL growth phase (14 UTC (7a PT)–smallest circles; 20 UTC (13 PT)–largest circles), while colors represent the BL wind direction. For each panel, the number of days used to generate a scatter plot ($n_{days}$) and the number of days within the sample that coincided with $O_3$ exceedance ($n_e$) is reported. The relatively small number of days compared to the analysis period (28 days) is due to selecting only days that yielded 4 or more extremum during the BL growth phase. Single variables (Figure 13a-e) typically had more days to produce a scatter plot compared to variable pairings (Figure 13f-i). The lack of days identified for variable pairings prevents a robust statistical analysis, but we can still comment on the distribution of the scatter.

The size of scatter points for all panels in Figure 13 generally increases along the y-axis, which is expected since we isolated a portion of the BL representing the growth phase. Northerly winds observed in some of the scatter points occurred near sunrise when the BL was still shallow. A clockwise rotation of the wind from northerly (and easterly) to southwesterly is observed with respect to time, and is in agreement with the evolution of BL winds in Figure 4l. The transition to southerly/southwesterly flows in the afternoon as the BL reached maximum height was likely a result of an increased thermal gradient across the coast into the afternoon accompanied by increased winds (Figure 4k and Figure 7a) confined within the BL (Figure 7b) in the form of onshore flow (i.e., a SB). Mesoscale wind transitions can complicate convective BL development and lead to challenges in separating buoyant and mechanical forcing caused by daytime heating and the SB. Furthermore, day-to-day differences in meteorological conditions and initial concentrations of $O_3$ and $O_3$ precursors at the start of the growth phase contributes to the time evolution of measurements at micrometeorological time-scales. As a result, the scatter that is produced for single variables (Figure 13a-e) is quite dispersed, especially in Figures 13d-e. Figures 13a-c, on the other hand, show some indication of a consistent positive trend that we will now explore below.

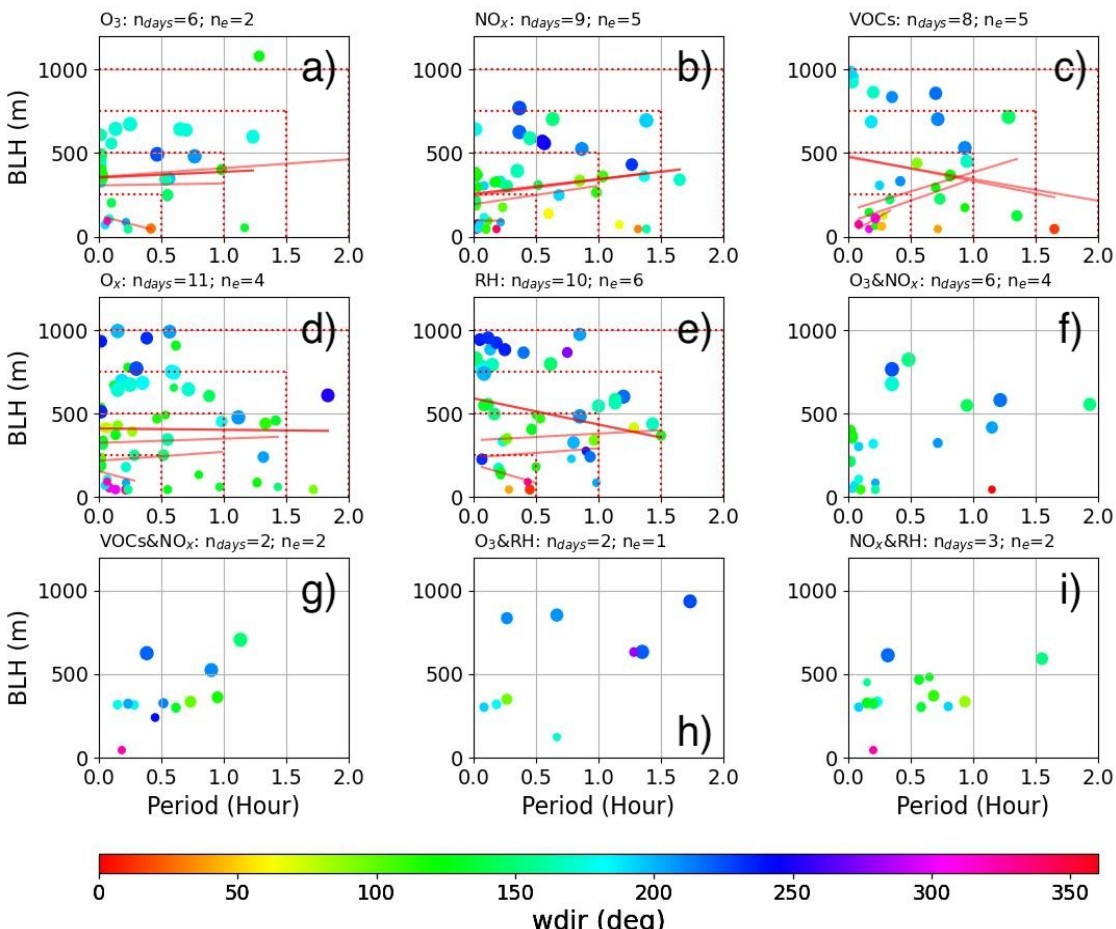

**Figure 13.** The temporal width of extremum ($\tau_{max}$–x-axis) versus BL height (y-axis) for a) $O_3$, b) $NO_x$, c) VOCs, d) $O_x$, e) surface relative humidity, f) $O_3$&$NO_x$, g) VOCs&$NO_x$, h) $O_3$&relative humidity, and i) $NO_x$&relative humidity. The size of markers depends on time, with the smallest markers coinciding near sunrise at 14 UTC and the largest markers coinciding with times near 20 UTC. The markers are color-coded by wind direction across the BL. Each panel includes a subtitle with the number of days ($n_{days}$) and the number of $O_3$ exceedance days ($n_e$). The red dashed rectangles in a)-e) represent thresholds in BL height and $\tau_{max}$, while lines of best fit, also in red, are applied against scatter grouped within thresholds.

The scatter points in Figures 13a-e are grouped within red dashed rectangles defined by increasing the BL and $\tau_{max}$ thresholds (i.e., excluding scatter that exceeds a threshold). The $\tau_{max}$ threshold is increased proportionately such that $\tau_{max}$ increases by 0.5 hours by every 250 m increase in the BL height threshold. Separate lines of best fit in red are overlaid across the scatter in Figures 13a-e for each threshold, and can be distinguished by noting that lines of best fit are contained within threshold rectangles. $O_3$ and $NO_x$ show a nearly consistent positive trend, while VOCs show a positive trend until larger thresholds are reached (i.e., when including $\tau_{max} > 1.5$ hours and BL height $> 750$ m). Consistent line trends were not found in Figures 13d-e.

Figure 14 reports the correlation coefficient between $\tau_{max}$ and BL height, and the % of scatter analyzed for fits as a function of the BL height and $\tau_{max}$ thresholds (x-axes). The $\tau_{max}$ determined by extrema and BL height is uncorrelated or weakly anti-correlated when the percentage of cases selected is high (i.e., higher thresholds) for relative humidity, $O_3$, and $O_x$; weakly correlated regardless of the threshold selected in the case of $NO_x$; and moderately correlated when at least 75% of cases are selected for VOCs.

The relatively large correlation coefficient when 3/4 of the scatter is selected for VOCs stands out compared to other variables, and points to a stronger relationship between the fine-scale variability of VOC measurements compared to other measurements as it relates to an increase in $\tau_{max}$ during the BL growth phase. However, when the thresholds are increased to include deeper BL heights (>750 m) and longer time-scales (>1.5 hours), then the correlation coefficient drops (Figure 14) and slopes reverse (Figure 13c) as a result of including cases where small $\tau_{max}$ coincides with deeper BL heights and large $\tau_{max}$ coincides with a shallower BL. The fringe cases occur around sunrise when winds are northerly or easterly, and during the end of the BL growth phase when onshore flow dominates the BL wind profile. These BL transitions can complicate the fine structure variability, which can lead to differences in the short-term time evolution of measurements that are believed to be responsible, at least in part, the increased dispersal of scatter points when increasing the BL height and $\tau_{max}$ threshold above 750 m and 1.5 hours, respectively, for VOCs.

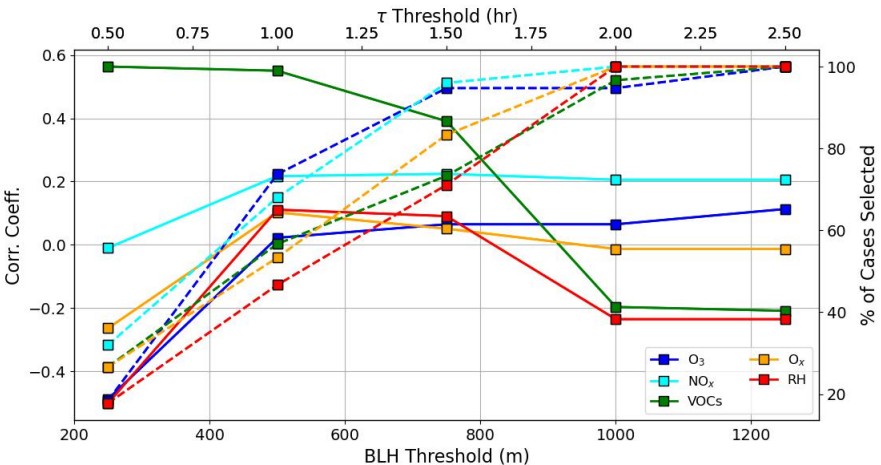

**Figure 14.** Correlation coefficients (solid lines) and % of scatter selected (dashed lines) for different BL and and $\tau_{max}$ thresholds (x-axes) color-coded by variables as reported in the plot legend.

The scatter in Figures 13f-i represent variable pairings that have a reduced sample size compared to single variables. When $O_3$ is paired with $NO_x$ or relative humidity, the scatter is dispersed and does not follow an obvious trend (Figure 13f, h). However, when $NO_x$ is paired with VOCs (Figure 13g) or relatively humidity (Figure 13i), the scatter visually appears more correlated. As mentioned above, $NO_x$ by itself was weakly correlated (Figure 13b), relative humidity was uncorrelated (Figure 13e), and VOCs was moderately correlated when extreme cases were excluded (Figure 13c). Combining these different

variables, regardless of whether single variables were correlated or not, results in a more positively correlated relationship. The increase in $\tau_{max}$ with respect to BL height is more obvious when variables are paired with other variables that tended to also exhibit increased $\tau_{max}$ with BL height (i.e., Figure 9 and 10). However, as mentioned above, the number of days where variable pairings exhibited four or more extremum was considerably less than single variables, thus weakening the strength of interpreting the results from Figure 13f-i.

## 6 Conclusions

In this study, we presented data collected during the Southwest Urban $NO_x$ and VOCs Experiment (SUNVEx) to understand the role of multi-scaled dynamics on air quality measurements in Pasadena, CA during August 2021. More than half the days experienced an $O_3$ exceedance event that coincided with elevated $PM_{2.5}$, increased $NO_x$ during nights preceding $O_3$ exceedance, and increased VOCs into the afternoon. The meteorological conditions averaged over $O_3$ exceedance days in Figure 4 favored increased temperature, reduced relative humidity, increased surface pressure, and reduced BL height and BL-averaged winds compared to non-exceedance days. Figures 2 and 3 confirm most of these findings with the exception of surface pressure and BL height, which did not always conform to averages conducted between $O_3$ exceedance and non-exceedance days (also indicated by correlation coefficients in Table 1). Evenings preceding $O_3$ exccedance days were typically more stable and stagnant compared to non-exceedance days as supported by shallower BL heights, reduced winds, and reduced surface temperature. The build-up of $NO_x$ and VOCs in Pasadena could have been caused by relatively stagnant conditions developing during evenings that preceded $O_3$ exceedance. However, the wind direction was considerably different during evenings leading up to O3 exceedance (northerly) and non-exceedance (southerly) days. Sources of biogenic VOCs or lingering smoke from wildfires could have been advected from the north during evenings preceding exceedance days, while a combination of urban emissions and marine air could have been advected in from the south during evenings preceding non-exceedance days. While the wind direction was markedly different at night between days when $O_3$ exceeded 70 ppb (northerly to northeasterly) versus days when $O_3$ did not exceed 70 ppb (southerly to southeasterly), the wind direction converged to southwesterly during the daytime, which typically coincided with increased wind speed as marine BL air propagated inland. An interesting meteorological feature worth noting was the semi-diurnal pressure pattern, whose troughs lined up near transitional periods (sunset and sunrise).

Superimposed on the broad air quality diurnal trends were fine structure details in the measurements believed to be partly related to the BL dynamics. To investigate the role of BL dynamics, 16 August, 2021 was chosen as a case study to evaluate details of the fine structure of air quality measurements from sunrise to the afternoon with the arrival of a SB. The winds across the BL during 16 August were relatively weak leading up to sunrise, with a predominately easterly flow in the evening near the surface that gave way to southwesterly flows into the afternoon. A diurnal pattern in wind direction above the BL featured northerly flows during the night that transitioned to easterly flows during the day. Winds descended into the Pasadena area between the early morning hours preceding sunrise and the time that the SB was first observed (near 18 UTC or 11a PT) as supported by Figures 7 and C1. The winds that descended before sunrise coincided with an increase in surface pressure (one of the peaks in the semi-diurnal pressure pattern) and a large dew point depression above the near-surface inversion observed

in nearby soundings. Increases in surface pressure, a large dew point depression, and a descending wind maximum supports general subsidence across the region. Interestingly, the HRRR failed to capture the descending wind jets despite accurately forecasting BL height, and the timing and intensity of the SB. The wind jets observed by the DL above the residual layer and BL were between 500 m and 1 km in thickness, occurred over several hours, and in terms of a descent into Pasadena, extended over a distance that likely exceeded 30 km (Appendix C). It is believed that descending winds supported adiabatic

compression, which enhanced the strength of the inversion and resulted in the decoupling and acceleration of winds above the inversion. Increased wind shear across the BL likely encouraged entrainment via mechanical coupling, which would explain the intermittency of enhanced wind jets. Studies have noted the importance of strong shear across the BL as a mechanism for enhanced entrainment (e.g., Fedorovich and Conzemius, 2008). The second of the two wind jets occurred as the BL deepened and a SB arrived into Pasadena, and coincided with a series of wind speed bursts and relatively strong updrafts that matched

with temporal fluctuations in $O_3$ and $NO_x$.

        In order to quantitatively address the precise role of BL dynamics on air quality measurements for the August 16[th] case study, a method was developed to isolate the fine structure variability of meteorological and air quality measurements from the diurnal trend. Scaleograms were created to understand the spectral characteristics of the local temporal variability within variable time series, while a Multivariate Spectral Coherence Mapping (MSCM) technique was developed and used to combine the maximum

normalized power spectrum density (PSD) from different variables to understand variable interdependencies. The fine structure details of measurements in Pasadena (air quality and dynamics) were most pronounced during BL transitions (i.e., evening transition, at sunrise and throughout the BL growth phase, and during the arrival of the SB). However, these measurements did not respond uniformly as the BL evolved. For instance, fine structure variability of chemical concentrations were absent during the evening transition and throughout the night, while certain meteorological measurements such as relative humidity,

and BL-averaged wind speed and wind speed shear varied considerably during those same periods. Most of the observations at Pasadena exhibited rapid changes during sunrise that ranged from the order of minutes to about an hour. The higher temporal resolution of $NO_x$ and VOC measurements enabled higher frequencies to be examined during BL growth. The temporal widths of extrema in $NO_x$ and VOCs increased as the BL deepened, which is corroborated against variations in the BL height with respect to time that ranged from 15 minutes shortly after sunrise to 1.5 hours as the BL climaxed. Other measurements did not

feature increases in the temporal width of extrema as the BL deepened (i.e., $\tau_{max}$ did not increase as the BL deepened). The arrival of the SB coincided with fine structure variations in $O_3$, $NO_x$, and $O_x$ that was well correlated. The temporal variability reduced from 30 minutes to about 15 minutes, which approximately matches the temporal spacing between relatively strong updrafts observed in Figure 7. An analysis conducted in Appendix D utilized the time-varying frequency structure of $O_3$, $NO_x$, and $O_3$&$NO_x$ scaleograms, and the linear increase in BL-averaged winds spanning sunrise to the passage of the SB to quantify

the intrinsic characteristics of observed oscillations in the data as well as mechanisms for their generation. A phase shift of 180 degrees observed between O3 and $NO_x$ oscillations during the SB is less clear. However, if $NO_x$ is somehow replenished by a nearby source or the reactivity time-scales of $NO_x$ are considerably longer than dynamical time-scales, then the variations would be dominated by the transport dynamics. Under these assumptions, the 180-degree difference between $O_3$ and $NO_x$

would therefore be related to an opposing profile structure (one quantity increases with height while the other decreases with height) subjected to the same dynamics that lead to vertical mixing via turbulent eddies.

Combining two variables using the MSCM technique highlighted the covariability of $O_3$ and $NO_x$ during the passage of the SB, while strong covariability was observed during the BL growth phase between $NO_x$, VOCs, and variations in BL height (Figures 9 and 10). Since the temporal width increased with respect to the BL deepening, we conducted a series of powerlaw fits to estimate the analytical structure of $\tau_{max}$ and its relation to BL height as justified by the nearly identical values between the average rate the BL height deepened ($dz_{BL}/dt$) and the slopes derived from fitting BL height to $\tau_{max}$ ($V_{\tau_{max}}$) normalized by $2\pi$. The motivation to normalize by $2\pi$ stemmed from analyses of $\tau_{max}$ plotted in Figure 11 and the separation of subsequent peaks (and troughs) highlighted in Figure 12. While modeling the BL height using Eq. (15) for different variables and variable pairings led to reasonable agreement to the observed BL height, it was found that $\tau_{max}$ from $NO_x$ resulted in a substantial overestimation of BL height, while $\tau_{max}$ derived from VOCs and VOCs&$NO_x$ slightly overestimated the BL height until 18.5 UTC (11:30a PT) as the modeled and observed behavior diverged significantly during the arrival of the SB. When the variations in BL height (not the overall trend in BL height) were combined with $NO_x$ or VOCs, the modeling of the BL height improved significantly. This shows that despite a good correlation between VOCs, $NO_x$, and variations in BL height, that the temporal structure observed in pollutants cannot be explained solely by BL dynamics since chemical concentrations (and therefore chemical reaction pathways) are altered in response to an evolving BL and as a result of advection. We must stress that the modeling exercise conducted should be viewed more as a thought experiment using some key assumptions in order to see if the fine structure details within the measurements scaled with BL growth. We did not consider advection in our analysis since we only had one set of measurements and not a network of observations. Furthermore, the derivation in Appendix B makes no assumptions about the time-scales used since this was a data-driven exercise.

Following the case study was an analysis of the entire August 2021 dataset in order to examine whether the case described above was unique or representative. This was done by isolating all days for each variable and variable pairing that had at least four temporal extrema isolated within scaleograms during the BL growth phase for each day. $O_3$, $NO_x$, and VOCs stood out as being most impacted by BL growth as a result of increased temporal width of extrema as the BL height increased. Other variables that consisted of 5 or more days did not feature as strong of a relationship. The scatter observed along the x-axis and y-axis for $O_3$, $NO_x$, and VOCs typically occurred during northerly flow regimes under weak wind conditions and a transition to southerly-to-southwesterly winds, respectively. Wind direction for variables that yielded better correlations between $\tau_{max}$ and BL height tended to feature a scatter that transitioned from northerly to southerly with respect to time as the BL deepened, though significant variability in wind direction was still observed. The case study presented was unique, but it can be argued that variable forcing conditions and reactions between a myriad of chemical species can lead to a unique set of conditions each day as well, which depend not only on the complex evolution between large-scale, mesoscale, and local (urban heat and wind island effects), but also from stationary and transiting sources of $O_3$ precursors that make make reproducibility challenging. However, we believe that the results enclosed present a promising method at disentangling the role of dynamics and chemistry on air quality evolution, which we hope to apply to a more continuous dataset in the future that features dynamics measurements at a higher temporal resolution.

The MSCM technique shows promise for applications where many potentially interrelated variables share spectral similarity at certain instances in space or time. The maximum normalization of the PSD, the overlapping of spectral characteristics displayed by scaleograms, the sorting of scaleograms, and the ordering of variables according to spectral similarity relative to a reference variable as outlined in Appendix A produces a ranking system that allows variables to be grouped and sorted. Quantitative evaluations of the spectral characteristics of measurements during BL transitions was made possible using the MCSM technique for an arbitrary number of variables (we examined combinations of 2 and 3 variables for this study). It is the hope to apply this technique to other field sites and with a network of observations to examine spectral shifting related to advective processes, provided that the spectral structure is relatively preserved between measurement sites.

## 6.1 Limitations and Path Forward

A major limitation in this work was the vertical and temporal resolution of the DL. Although we were able to link the dynamics with air quality measurements, the rapid variability in air quality measurements was often at a temporal scale much finer than the 15-minute scan cycle of the DL. Based on the 16 August case study, we believe that a time resolution closer to in situ chemistry measurements would have resulted in stronger relationships between BL-averaged winds and chemistry measurements given the apparent covariability observed between measurements seen in Figure 7, for example. While the vertical resolution of the DL was relatively high (20 m near the surface), a resolution of 20 m across the depth of the profile from a DL or tower measurements with spacing on the order of 5 m near the surface may have improved the diagnosis of the BL height during transitional time periods (i.e., evening and morning transition). The relatively coarse temporal and vertical resolution sometimes led to a discontinuous structure that was not always easy to process via EMD, especially for the BL height product.

A second limitation of the dataset was the relatively short duration over which measurements were taken at Pasadena – only a monthlong dataset. A more continuous record featuring a high temporal resolution dataset (i.e., 60 s time resolution) that spans years so that the same month is covered multiple times will allow stronger statistical relationships to be developed between dynamics and air quality measurements, provided that the resolution of the DL (temporal and vertical) is also improved.

A third limitation stems from the analysis being conducted at a single site–Eulerian view. As a result, advection could not be evaluated and characterized in the context of the fine structure details of measurements. It would be of interest to expand this analysis to an observational site featuring multiple measurement platforms at different spatial distances as well as within different urban centers.

While studies have illustrated the micrometeorological role of BL transitions and SBs on air quality measurements, this study presented a quantitative analysis of the temporal structure observed in both air quality and meteorological measurements as well as the degree to which different variables are correlated. The techniques developed not only led to the characterization of the fine structure variability with respect to BL growth, but an analysis of the intrinsic characteristics of wave-like features observed in measurements during the propagation of a SB. A logical next step would be to extend this analysis with a collocated $O_3$ lidar to link what is observed at the surface (as done in this study) with $O_3$ concentrations aloft. This would allow not only

a separation of local versus non-local $O_3$ events, but would allow a link between interactions between pollutants in the BL with pollutants in the free troposphere through entrainment.

*Data availability.* The data used for this analysis can be found in the Chemical Sciences Laboratory website under (Brewer, 2021) and (Brown, 2021) for the stationary Doppler lidar and in situ chemistry, respectively, at Pasadena, California.

## Appendix A: Ranking Variables With Shared Spectral Characteristics Relative to a Chosen Reference Variable

Suppose we are interested in determining two variables that share similar scaleogram characteristics. We can start with Eq. (8), set $L = 2$ since we are only interested in two variables out of $M$ possible variables, define our reference variable by index $m_0$, and sort through the remaining $M - 1$ variables to examine spectral similarities. Eq. (A1) represents the maximum normalized spectral product between reference variable, $x_{m_0}$, and the variable that results in maximum spectral coherence, i.e., $x_{m_1}$,

$$\widetilde{C}^2_{m_1} = \left(\widetilde{C}^1_{m_0} \widetilde{\chi}^{m_1}_\psi\right)^{\frac{1}{2}} \tag{A1}$$

where $\widetilde{C}^1_{m_0} = \widetilde{\chi}^{m_0}_\psi$ by virtue of Eq. (8). The operation in Eq. (A1) results in two variable subspaces: a subspace comprised of the reference variable and the variable selected based on maximum spectral coherence, $\boldsymbol{Y}_1$, and a variable subspace that consists of leftover variables that can be evaluated when maximizing spectral coherence for higher order moments, $\boldsymbol{X}_1$, i.e.,

$$\boldsymbol{Y}_1 = \{x_{m_0}, x_{m_1}\} \tag{A2}$$

and

$$\boldsymbol{X}_1 = \{x_{n_1}, x_{n_2}, ..., x_{n_{M-2}}\} \tag{A3}$$

Here, $\boldsymbol{X}_1 \cap \boldsymbol{Y}_1 = 0$, $\boldsymbol{X}_1 \oplus \boldsymbol{Y}_1 = \boldsymbol{X}$, $\{\boldsymbol{X}_1, \boldsymbol{Y}_1\} \subseteq \boldsymbol{X}$, $\boldsymbol{X} = \{x_{m_0}, x_{m_1}, ..., x_{M_m}\}$, and subscript, $n$, is a dummy placeholder to denote the remaining $M - 2$ variables that can be selected if we were to maximize spectral coherence for a third moment calculation. It should be noted that index, $m_1$, could pertain to any variable within the list of available variables (i.e., $\boldsymbol{X}_0 = \{x_{n_1}, x_{n_2}, ..., x_{n_{M-1}}\}$), where $x_{m_0} \notin \boldsymbol{X}_0$.

If we take the result from Eq. (A1) and sort through the remaining variables in Eq. (A3), we can isolate the variable that would maximize spectral coherence for a third moment calculation ($L = 3$), i.e.,

$$\widetilde{C}^3_{m_2} = \left(\widetilde{C}^2_{m_1}\right)^{\frac{2}{3}} \left(\widetilde{\chi}^{m_2}_\psi\right)^{\frac{1}{3}} \to \left(\widetilde{C}^1_{m_0} \widetilde{\chi}^{m_1}_\psi \widetilde{\chi}^{m_2}_\psi\right)^{\frac{1}{3}} \tag{A4}$$

thus leading to a reduced variable subspace of remaining variables that can be selected, i.e., $\boldsymbol{X}_2$, and an inflated variable subspace comprised of selected variables, $\boldsymbol{Y}_2$, i.e.,

$$\boldsymbol{X}_2 = \{x_{n_1}, x_{n_2}, ..., x_{n_{M-3}}\} \tag{A5}$$

and

$$\boldsymbol{Y}_2 = \{x_{m_0}, x_{m_1}, x_{m_2}\} \tag{A6}$$

As before, the following conditions apply, but this time to Equations (A5) and (A6): i.e., $\boldsymbol{X}_2 \cap \boldsymbol{Y}_2 = 0$, $\boldsymbol{X}_2 \oplus \boldsymbol{Y}_2 = \boldsymbol{X}$, and $\{\boldsymbol{X}_2, \boldsymbol{Y}_2\} \subseteq \boldsymbol{X}$.

Following this progression, we can define a fourth moment (i.e., $L = 4$) along with the following variable subspaces, i.e.,

$$\widetilde{C}_{m_3}^4 = \left(\widetilde{C}_{m_2}^3\right)^{\frac{3}{4}} \left(\widetilde{\chi}_\psi^{m_3}\right)^{\frac{1}{4}} \rightarrow \left(\widetilde{C}_{m_0}^1 \widetilde{\chi}_\psi^{m_1} \widetilde{\chi}_\psi^{m_2} \widetilde{\chi}_\psi^{m_3}\right)^{\frac{1}{4}} \tag{A7}$$

$$\boldsymbol{X}_3 = \{x_{n_1}, x_{n_2}, ..., x_{n_{M-4}}\} \tag{A8}$$

$$\boldsymbol{Y}_3 = \{x_{m_0}, x_{m_1}, x_{m_2}, x_{m_3}\} \tag{A9}$$

As can be seen, the sequence of operations outlined above can be extended into general terms, i.e.,

$$\widetilde{C}^L(\tau, b) = \left(\prod_{j=1}^{L} \widetilde{\chi}_\psi^j(\tau, b)\right)^{1/L} \tag{A10}$$

where the following conditions are satisfied regardless of the moment order, i.e.,

$$\{\boldsymbol{X}_0 \cap \boldsymbol{Y}_0, \boldsymbol{X}_1 \cap \boldsymbol{Y}_1, ..., \boldsymbol{X}_M \cap \boldsymbol{Y}_M\} = 0 \tag{A11}$$

$$\{\boldsymbol{X}_0 \oplus \boldsymbol{Y}_0, \boldsymbol{X}_1 \oplus \boldsymbol{Y}_1, ..., \boldsymbol{X}_M \oplus \boldsymbol{Y}_M\} = \boldsymbol{X} \tag{A12}$$

Lastly, as the moment order approaches the number of available variables, the inflated variable subspace converges to $\boldsymbol{X}$, i.e.,

$$\lim_{L \to M} \boldsymbol{Y}_L = \boldsymbol{X} \tag{A13}$$

The additional benefit of the approach outlined above is that the variables are automatically ordered based on the degree of spectral similarity with the reference variable regardless of the number of variables used in the operational sequence, thus leading to the ranking of variables with respect to an arbitrary reference variable.

## Appendix B: Derivation of BL Height Using Maximum Periods Determined from Scaleograms

Instead of equating the time derivative of the BL height to Eq. (12) as was done in Eq. (13), let us take the time derivative of Eq. (12) directly, i.e.,

$$\frac{dV_{\tau_{max}}}{dt} = \frac{d}{dt}\left(\frac{z_{BL}}{\tau_{max}}\right) \rightarrow \frac{1}{\tau_{max}} \frac{dz_{BL}}{dt} - \frac{z_{BL}}{\tau_{max}^2} \frac{d\tau_{max}}{dt} \tag{B1}$$

From there, we can divide Eq. (B1) by $V_{\tau_{max}}$

$$\frac{d\ln V_{\tau_{max}}}{dt} = \frac{1}{V_{\tau_{max}}\tau_{max}} \frac{dz_{BL}}{dt} - \frac{z_{BL}}{V_{\tau_{max}}\tau_{max}^2} \frac{d\tau_{max}}{dt} \tag{B2}$$

and substitute $V_{\tau_{max}}$ for $z_{BL}/\tau_{max}$ into the right hand side of Eq. (B2), followed by a series of cancellations, i.e.,

$$\frac{d\ln V_{\tau_{max}}}{dt} = \frac{d\ln z_{BL}}{dt} - \frac{d\ln \tau_{max}}{dt} \tag{B3}$$

The remaining terms can be merged with the time derivative of variables to simplify the form into the time derivative of the natural log of variables (Eq. (B3)). We can now integrate Eq. (B3) with respect to time, which leads to

$$\int_{\ln V_{\tau_{max}}(t_0)}^{\ln V_{\tau_{max}}(t)} d\ln V_{\tau_{max}} = \int_{\ln z_{BL}(t_0)}^{\ln z_{BL}(t)} d\ln z_{BL} - \int_{\ln \tau_{max}(t_0)}^{\ln \tau_{max}(t)} d\ln \tau_{max} \tag{B4}$$

where $t_0$ refers to the time at sunrise while $t$ is some arbitrary time during the growth phase of the BL. Carrying out the integration in Eq. (B4) leads to

$$\ln\left(\frac{V_{\tau_{max}}(t)}{V_{\tau_{max}}(t_0)}\right) = \ln\left(\frac{z_{BL}(t)}{z_{BL}(t_0)}\right) - \ln\left(\frac{\tau_{max}(t)}{\tau_{max}(t_0)}\right) \tag{B5}$$

Taking the exponential of Eq. (B5) and subsituting in $V_{\tau_{max}}(t)=z_{BL}(t)/\tau_{max}(t)$ and $V_{\tau_{max}}(t_0)=z_{BL}(t_0)/\tau_{max}(t_0)$ results in

$$\frac{z_{BL}(t)}{\tau_{max}(t)}\frac{\tau_{max}(t_0)}{z_{BL}(t_0)} = \frac{z_{BL}(t)}{z_{BL}(t_0)} - \frac{\tau_{max}(t)}{\tau_{max}(t_0)} \tag{B6}$$

which can then be manipulated to isolate $z_{BL}(t)$, i.e.,

$$z_{BL}(t) = z_{BL}(t_0)\frac{\tau_{max}(t)}{\tau_{max}(t_0)}\left(1 - \frac{\tau_{max}(t_0)}{\tau_{max}(t)}\right)^{-1} \tag{B7}$$

A major shortcoming of Eq. (B7) is the blow-up that occurs at sunrise when $t=t_0$. However, if we recall that the fits to derive slopes in Figure 11b were done against data that was well correlated, and that the slope suggests a nearly constant velocity, then we can assume that $dV_{\tau_{max}}/dt=0$ such that Eq. (B7) simplifies to

$$z_{BL}(t) \approx z_{BL,0}\frac{\tau_{max}(t)}{\tau_{max,0}} \tag{B8}$$

Here, Eq. (B8) represents the analytical relation used to model BL heights with derived $\tau_{max}$'s in Figure 11c. Although a tedious exercise, Eq. (B8) assumes that the BL height can be modeled using changes in the temporal structure if the initial conditions at sunrise for both BL height and $\tau_{max}$ are known. Furthermore, the form of $\tau_{max}(t)$ is represented as a general function, but as determined by the variable and variable pairings selected for Figure 11, follows a powerlaw behavior. We

suspect that this form could be potentially used to separate the dynamical influence of BL growth from advection and chemical reactions that would also change the temporal characteristics of air quality measurements.

## Appendix C: Descending Wind Maxima: Estimating the Rate of Descent and Horizontal Scale of Wind Jets

Profiles of wind speed in Figure C1 during times intersecting gray arrows in Figure 7a-b showed a general descending pattern and intensification. The wind maximum observed before sunrise in Figure C1a intensified by about 2 m s$^{-1}$ and descended more than 300 m in 2 hours, yielding an average rate of descent of about 4 cm s$^{-1}$, which is in agreement with other studies that have investigated this region (e.g., Glendening et al., 1986; Lu and Turco, 1995). It is worth noting that an increase in surface pressure in Figure 7d occurred concomitantly with descending winds in Figure C1a. The development of a second wind maximum during the BL growth phase in Figure C1b was not as pronounced as Figure C1a. However, winds did intensify with respect to time while yielding an overall downward displacement of the bottom portion of enhanced winds above the BL (refer to slanted black line in Figure C1b). Unlike the time period spanning profiles in Figure C1a, an increase in surface pressure spanning the time period of profiles in Figure C1b was not observed. However, it must be kept in mind that surface heating and onshore flow will contribute to modifications in the mass column and therefore the surface pressure.

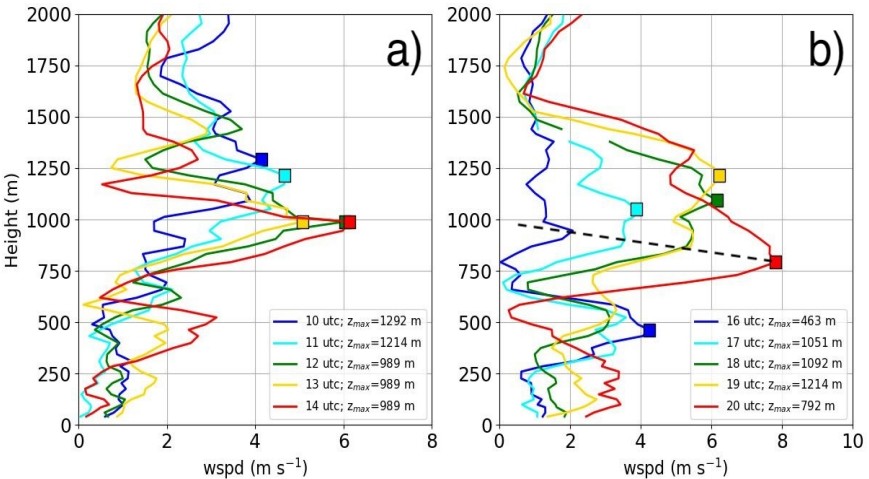

**Figure C1.** Profiles of wind speed with wind speed maximum plotted as squares above the BL during a) 10, 11, 12, 13, and 14 UTC, and b) 16, 17, 18, 19, and 20 UTC. Overlaid in b) is a dashed line that illustrates the general descent path of the bottom portion of enhanced winds above the BL.

The shallow wind jets observed above the residual layer and BL did not appear in HRRR time-height cross-sections (Figure 6b) despite reproducibility of the SB intensity and timing. The wind jets observed were narrow in depth, but sufficient enough to be resolved by the HRRR. To get a sense of the minimum length of the wind jet, several back-of-the-envelope calculations were made that combined the continuity equation and the strength of the descending wind maximum over a two hour period for profiles in Figure C1a.

The descending wind jet was observed for the first three profiles in Figure C1a. After the descent of the wind maximum, the jet was observed for two more hours with intermittent episodes of weakening and intensification that was likely influenced by shear-induced mixing or instabilities that temporarily disrupted the strength of the wind jet. We used the height of the

descending wind maximum between profiles 1 and 2 (blue and cyan profiles), and 2 and 3 (cyan and green profiles) to determine the descending rate, i.e.,

$$w_{12} = \frac{\delta z_{12}}{\delta t} = -\frac{1292m - 1214m}{3600s} = -0.022 ms^{-1} \tag{C1}$$

$$w_{23} = \frac{\delta z_{23}}{\delta t} = -\frac{1214m - 989m}{3600s} = -0.063 ms^{-1} \tag{C2}$$

where $\delta t$ is the time difference between profiles and $\delta z$ represents the height difference of the jet maximum between two different instances in time. As the wind jet descended, the horizontal winds intensified to a maximum after reaching the residual layer or the BL. The height where the wind jet leveled off in Figure C1a was approximately 1 km, which is similar to the BL height from the previous day. We suspect that the subsiding wind jet could not successfully penetrate the residual layer. It is believed that the downward motion converted into horizontal motion as the wind jet leveled horizontally, which led to enhancements from the north above the residual layer. Using this argument, we estimated the incremental distance traveled in the horizontal direction as a result of wind jet acceleration. We first started with the continuity equation in two dimensions, i.e.,

$$\nabla \cdot \boldsymbol{v} = \frac{\partial v}{\partial x} + \frac{\partial w}{\partial z} \rightarrow \frac{\delta v}{\delta x} + \frac{w_{12} - w_{23}}{\overline{z}_{12} - \overline{z}_{23}} = 0 \tag{C3}$$

where $\overline{z}_{12}$ and $\overline{z}_{23}$ represent the average between layers for which $w_{12}$ and $w_{23}$ were calculated, $\delta v$ as the difference in the horizontal wind between profiles averaged from 1 to 2 (blue and cyan) and from 2 to 3 (cyan to green), i.e., $\delta v = \overline{v}_{12} - \overline{v}_{23}$, and $\delta x$, the distance increment to be calculated. Rearranging Equation (C3) and plugging in numbers leads to the additional distance winds traveled during descent and acceleration, i.e.,

$$\delta x = -\frac{\overline{v}_{12} - \overline{v}_{23}}{w_{12} - w_{23}} \left(\overline{z}_{12} - \overline{z}_{23}\right) = -\frac{4.23 ms^{-1} - 5.18 ms^{-1}}{-0.022 ms^{-1} + 0.063 ms^{-1}} \left(1253m - 1102m\right) \approx 3.55 km \tag{C4}$$

The total distance traveled can be estimated as

$$X = \int_{t_1}^{t_2} \overline{v}_{12} dt + \int_{t_2}^{t_3} \overline{v}_{23} dt = \overline{v}_{12} \int_{t_1}^{t_2} dt + \overline{v}_{23} \int_{t_2}^{t_3} dt \rightarrow \left(\overline{v}_{12} + \overline{v}_{23}\right) \delta t = 33.84 km \tag{C5}$$

where the acceleration of winds represents approximately 10% of the total distance the wind jet propagated in two hours, i.e., $\delta x/X \approx 10\%$. The persistence of the wind jet supports the minimum length-wise extent of 33.83 km, which represents a little more than 10 horizontal grid points within the HRRR (i.e., both the vertical an horizontal dimension of the wind jet are resolvable). The proximity to the San Gabriel Mountains and the northerly flow of the wind jet points to the likelihood of a downslope wind event. It is unclear how far this wind jet propagated southward. However, if the wind jet remained relatively unimpeded during the time that it was observed (i.e., 5 hours), then it is reasonable to assume that the distance the wind jet could have propagated was close to 80 km. The backdrop of the mountains can be seen in Pasadena, and the distance from Pasadena to the coast in a north-south line is a little more than 50 km. The total distance that could have been traversed by

910 the wind jet exceeded the distance from the mountains to the coast. Downslope winds propagating offshore are not unheard of (e.g., Strobach et al., 2018), and the implications for air quality are profound since biogenic sources of VOCs could mix in with marine BL air, thus complicating the distribution of $O_3$ precursors that impact the air quality outlook for the following day.

## Appendix D: Estimating the Scale and Intrinsic Frequency of 'Oscillations' Observed in $O_3$ and $NO_x$ During a SB
Transition

A remarkably coherent feature observed in the fine structure details of $O_3$ and $NO_x$ were temporal oscillations that increased in frequency from about a half hour to 15 minutes during the SB transition. Close inspection of the BL-averaged wind speed in Figure D1a supports a general linear increasing trend from sunrise into the afternoon as the BL climaxed. The decrease in temporal width coincided with increased BL-averaged winds with respect to time. We hypothesize that increased winds with
920 respect to time during the SB transition led to swifter advection across the region that manifested as a decrease in the temporal width of extremum (sort of like a compressed wave). For coherent wave-like structures that do not to change with space or time

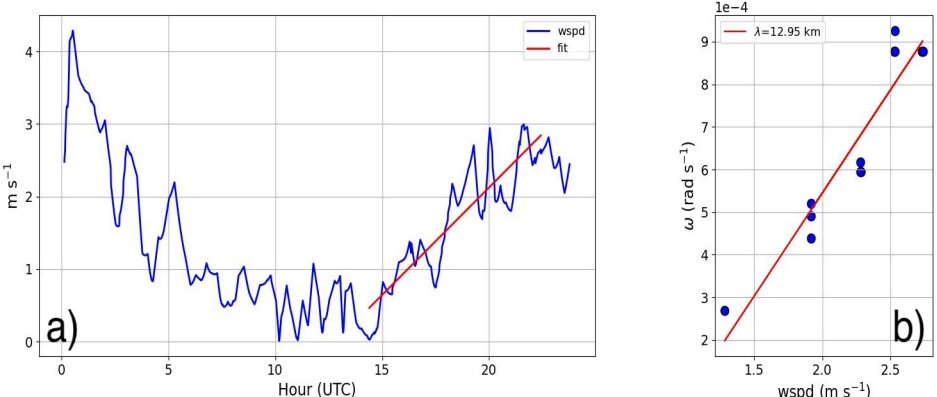

**Figure D1.** a) BL-average wind speed (blue) overlaid with a line of best fit extending from sunrise to the maximum height of the BL (red) and b) a scatter plot between derived apparent frequency, $\omega$, and BL-averaged winds derived from the line of best fit in a). The line of best fit b) was used to derive the wavelength, $\lambda$ and intrinsic frequency, $\hat{\omega}$.

– as assumed in our case – we can relate the intrinsic frequency of the oscillation with the apparent (or observed) frequency of the oscillation (i.e., from in situ measurements) by considering the background winds in which the oscillation was embedded, i.e.,

$\hat{\mathbf{c}} = \mathbf{c} - \mathbf{U}$ (D1)

where $\hat{\mathbf{c}}$ is the intrinsic phase speed, $\mathbf{c}$ is the apparent phase speed, and $\mathbf{U}$ is the horizontal wind vector. The phase speeds are defined by the frequency ($\hat{\omega}$ and $\omega$) and wavenumber within the horizontal plane in the direction of propagation, $\kappa$, i.e., $\hat{c} = \hat{\omega}/\kappa$

and $c=\omega/\kappa$. Substituting the definition of intrinsic and apparent phase speed into Equation (D1) and dotting $\kappa$ into the Equation (D1) with phase speeds expressed in terms of frequency and wavenumber leads to

$$\hat{\omega} = \omega - \kappa \cdot \mathbf{U} \tag{D2}$$

where $\omega$ is related to the temporal extrema observed in $NO_x$ and $O_3$ scaleograms (most clearly demonstrated in Figure 1).

Revisiting the assumption that the intrinsic characteristics of the wave-like feature does not change with time, i.e., $\hat{\omega}(t)=constant$, we can take the time derivative of Equation (D2), which results in a relationship between how the apparent frequency of oscillations and the wind changes with time, i.e.,

$$\dot{\omega} - \kappa \cdot \dot{\mathbf{U}} = \mathbf{0} \tag{D3}$$

where the time derivative of $\hat{\omega}$ is zero since $\hat{\omega}$ is assumed to be constant in time. This results in a relationship between the apparent frequency and the horizontal wind with the wavenumber, $\kappa$. The apparent frequency is related to the temporal widths derived from scaleograms (e.g., Figure 1b); however, since temporal widths define only a portion of the oscillation, then $\tau_{max}$ cannot represent the true period. This can be understood by comparing the time difference between subsequent peaks (or troughs) in Figure 1b to the temporal widths defined in Figure 1d. For instance, the time difference between subsequent peaks (and troughs) after 17 UTC ranges from 2 to 3.5 hours, while the the widths defined in Figure 1d are generally less than a half hour. Dividing the average time difference between subsequent peaks (or troughs) with the average $\tau_{max}$ approximately results in $2\pi$, i.e., $\overline{\tau}/\overline{\tau}_{max} \approx 2\pi$. Therefore, we can relate $\tau_{max}$ to $\omega$ in the following way, i.e.,

$$\omega = 2\pi f \rightarrow \frac{2\pi}{\tau} \approx \frac{1}{\tau_{max}} \tag{D4}$$

where $\omega$ is related to the regular frequency, $f$, while $f$ is simply the inverse of the period defining an oscillation, $\tau$. The ratio between $\tau$ and $\tau_{max}$ is approximately $2\pi$, and as such $\omega$ can simply be expressed as $1/\tau_{max}$.

Now that we have defined $\omega$ and $U$, we now attempt to determine the wavelength, $\lambda$, of the oscillation with the requisite assumption that the intrinsic structure of the oscillation did not change with respect to time. Figure D1b shows the BL-averaged wind speed derived from the line of best fit (i.e., red line in Figure D1a) during times coincident with the temporal widths of extrema versus $\omega$ for $O_3$, $NO_x$, and $O_3\&NO_x$. Through the scatter is a line of best fit with a positive slope and a correlation coefficient of about 0.95. The equation of the fit is represented as

$$\omega = \frac{\Delta\omega}{\Delta U} U + \omega_0 \tag{D5}$$

where $\frac{\Delta\omega}{\Delta U}$ represents the slope and $\omega_0$ is an intercept. Taking the time derivative of Equation (D5) leads to a form that is similar to Equation (D3), which can be used to solve for $\kappa$, i.e.,

$$\frac{\Delta\omega}{\Delta U} \rightarrow \frac{\dot{\omega}}{\dot{U}} = \kappa \tag{D6}$$

The slope (or wavenumber, $\kappa$) is $4.85^{-4}$ m$^{-1}$, which after rearranging $\kappa=2\pi/\lambda$ to solve for $\lambda$ leads to a wavelength of about 12.95 km. If we substitute Equation (D5) in Equation (D2), we find that $\hat{\omega}=\omega_0$. This suggests that the intercept can be used to

derive the intrinsic frequency. The intrinsic frequency in this case was negative (i.e., $\hat{\omega}$=-4.25*10$^{-4}$ rad s$^{-1}$) with an absolute period of about 4.1 hours (i.e., $2\pi/\hat{\omega}$). The intrinsic period is larger than periods observed from a stationary frame of reference

because winds in the BL assist in the propagation of an organized coherent wave-like pattern in the direction of the SB. The negative intrinsic frequency suggests that waves propagated in the opposite direction of the SB, which, given the derived intrinsic period and wavelength of the oscillation leads to an intrinsic phase velocity of -0.88 m s$^{-1}$. The winds in the BL are larger than the intrinsic phase speed of the assumed wave, and thus the wave-like pattern as observed at the surface from a stationary point of reference will propagate in the direction of the SB at an apparent phase speed less than winds from the SB.

Examining Figure 7a-b, it is clear that a relatively strong wind descending from above to the height of the BL as a SB propagated into Pasadena led to a complicated dynamical interaction that generated bursts of turbulence as well as wave-like features observed in in situ measurements of $O_3$ and $NO_x$. Furthermore, upon studying the wind profiles in Figure 7a near the top of the BL during the time the SB propagated into Pasadena, we see a wave-like oscillation with a period on the order of several hours. We surmise that relatively strong winds riding in a direction nearly opposite to SB propagation

initiated a wave-like response across the BL interface that was driven by the undercutting SB and strong winds descending from above. The negative intrinsic frequency suggests that any waves forced by this interaction occurred because of relatively stronger winds above the BL compared to winds from the SB, thus leading to an intrinsic phase velocity in a direction that was opposite of the SB. The strong directional shear extending from the surface to some height above the BL as a SB propagated inland provided ideal conditions for the generation of spanwise vorticity. Spanwise vorticity and mechanical lifting from a

propagating SB provided an ideal set of conditions that support the generation of gravity waves with a wavelength comparable to those generated by a SB in simulations featuring a similar horizontal wind profile and SB depth (Fovell, 2005). The unique approach to characterizing the wave-like dynamics of the fine structure as observed by in situ chemistry measurements was made possible with the development of techniques outlined in Section 2 of this paper.

*Author contributions.* EJS conceptualized, conducted the scientific analysis, and drafted the manuscript. SB and BJC helped with the focus

and writing of the manuscript, and participated in discussions on the analysis. SSB assisted with improved focus and messaging related to explanations tied to the chemistry. KZ improved the messaging and focus of the manuscript, suggested a deeper dive into in situ pressure measurements, and incorporated Section 2.1.2. MC assisted with the messaging and writing of manuscript, and incorporated Section 2.1.3. AWB, LX, YLP, and CES assisted with the messaging and writing of the manuscript. CW, JP, JG, BM, MH, and RM were involved in data acquisition and curation.

*Competing interests.* At least one of the (co-)authors is a member of the editorial board of Atmospheric Chemistry and Physics

*Acknowledgements.* The corresponding author would like to acknowledge the careful attention and involvement of co-authors, which led to significant improvements in the focus and messaging of the manuscript. In addition, the authors are grateful for the feedback from one anonymous reviewer and Ian Faloona that led to substantial improvements to the delivery and scientific analysis of the manuscript.

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
