# Peer review of "An Air Quality and Boundary Layer Dynamics Analysis of the Los Angeles Basin Area During the Southwest Urban NOx and VOCs Experiment (SUNVEx)"

_EGUsphere, 2024_

## Author Response (AR1)

Hi Ian Faloona,

Thank you for your interest in this research and your detailed evaluation of the manuscript. Your attention to detail highlighted areas where the manuscript could be improved. Below are separate sections that address the general and specific comments. I note where I have made changes in the manuscript and include plots to support claims within the manuscript. While I believe that I have successfully addressed comments and concerns, I welcome further dialogue if areas are still unclear.

Overall: The comments received focused on several key issues. First, section 3, which detailed the evolution of diurnal trends from in situ chemistry/meteorology and the DL measurements during the month of August, lacked a discussion comparing Figures 2 and 3 with Figure 4. This led to an overemphasis of Figure 4, and no discussion contrasting results between Figures 2 and 3 with Figure 4. Second, there wasn't a strong enough justification for the 2*pi normalization used to model BL heights (a new Figure 12 was added to support this discussion). Third, we did not discuss or consider the possible impacts of advection in section 4 or 5. Fourth, a stronger analysis of descending winds into Pasadena was needed (please see Appendix C). Fifth, the analysis in section 5 needed significant improvement in presentation quality and explanation. Sixth, a deeper analysis of the representativeness of HRRR in interpreting the winds into Pasadena was requested.

We have consolidated the two plots in section 5 (i.e., Figures 11 and 12) into a single plot that now includes wind direction (scatter color-coded by wind direction). This is now Figure 13. Also included in Section 5 is Figure 14, which gives a more quantitative assessment of scatter distributions in Figure 13. Included in Section 4 is an additional Figure that shows the performance of the HRRR (now Figure 6) along with a brief discussion comparing the Doppler lidar and HRRR. The new Figure 12 helps justify the 2*pi normalization. It is believed that we made significant strides to address these concerns which are outlined in the responses to general and specific comments below.

Thanks again.

General Comments:

Be specific about which BLH data is being shown and used in the analysis (HRRR is hourly, Doppler Lidar is 4/hr). If you are using the higher rate w-variance technique from the DL, then perhaps you could compare it with the HRRR data set to see if the model is doing a decent job of estimating ABLH (I would guess that it is not).

EJS: 1. I do allude to using "observations of BL height" in the Figure 3 caption. Also, you can assume that I am using DL-derived BL heights in lines 272-274. Line 439-440 makes it clear that I'm using BL height derived from observations in Figure 7. I make it clear when discussing the HRRR that the BL heights for Figure 5 are from the HRRR. I make it clear that the BL height used is from observations when discussing Figure 8 (e.g., lines 489-491). This should intuitively carry over for Figures 9 and 10 without being said explicitly. Since Figure 11 uses $\tau_{max}$, then it should be clear without stating explicitly that I'm using BL height derived from observations. However, I will make it clear in the lidar description section which products are used, including BL height. It should be emphasized that the HRRR was only used to characterize the regional flow during the case study with pm2.5 observations overlaid. Text has been added in lines 121-124 in the revised manuscript to make this clear: "In this study, the horizontal and vertical winds, and BL height from the DL are used to describe the evolving wind conditions at Pasadena spanning nocturnal and daytime periods during the month of August. Additionally, departures in the mean wind and BL structure as

observed by the DL are examined when addressing the fine-scale variability reported in scaleograms (discussed in the Methods subsection). "

The machinations to develop a vertical velocity scale seems unnecessary: you can use the w-variance measurements to estimate w* from convective similarity theory.

EJS: The point was not to develop a vertical velocity scale; rather, the point was to see whether temporal fluctuations of surface measurements from morning to afternoon could be linked to or is correlated to the growth of the boundary layer. The scaleograms, especially for NOx and VOCs for the case examined, showed these measurements changing abruptly at sunrise followed by a transition to increasingly broader temporal variations as the BL height increased in depth. The idea was to combine the BL height coincident in time with peaks in a scaleogram to determine whether the slope of the result agreed with how the BL height changed with time. This was a data-driven exercise, and there were no assumptions made about time-scales, only that time-scales derived from scaleograms increased with increasing BL height. This motivated the discussion and presentation of Figure 10 (now Figure 11) shown later. References to velocity scales have been changed to BL height growth rate to avoid confusion.

Throughout solar noon is identified as 19 UTC, but solar noon is much closer to PST, not PDT, and is therefore more like 20 UTC. Also note that the length of the day decreases by about 1 hour across the month of August in SoCal.

EJS: You are correct. I have changed to noon.

The exact arrival time of the sea breeze should be better established. I would suggest plotting dew point temperature (or water mixing ratio) which are not directly influenced by air temperature to most clearly indicate the arrival of the marine layer into Pasadena. See Mayor (2011) and Wang & Ullrich (2018). This should be included in Figure 6.

EJS: I have removed Figure 6c, shifted 6d into the place of 6c, and plotted a new Figure 6d that includes temperature, dew point, pressure, and BL-averaged wind speed (note that Figure 6 is now Figure 7). I place more confidence in inferring the arrival of the seabreeze via the dynamics. Not much shows up in dew point until about 21 utc, which coincides with the transition to a more southwesterly flow. However, southerly flows develop well before 21 utc along with the generation of wind speed bursts. With a southerly flow, you still have a substantial onshore component leading up to Pasadena. The increase in dew point during a southwesterly flow occurs when a fetch between the coast and Pasadena is a minimum. This could be enough to carry air with an increase in dew point into the region, while increased fetch (southerly flow) allows a longer time for turbulent mixing to vertically distribute the moisture. Second, the HRRR shows onshore flow as early as 19 utc across much of the coastline. Lastly, studies (e.g., Banta et. al. 2005) have included analyses of relative humidity and winds to diagnose the timing of the baybreeze in Houston. A reduction in temperature would reduce the saturation vapor pressure and increase relative humidity if the mixing ratios remain constant. However, if the mixing ratio increases as well, this would show up as a potentially strong increase in relative humidity. Therefore, based on this argument as well as studies using relative humidity to diagnose seabreeze/baybreeze transitions, I stand by the use of relative humidity as a reasonable indicator. Another consideration to keep in mind is the differential heating that drives the seabreeze. Changes in forcing conditions throughout the day could lead to changes in the seabreeze propagation speed and

characteristics.  Studies have shown that density current-like mesoscale flows can penetrate and recede within a day depending on internal and external forcing conditions (Lareau and Clements 2015). Please see the caption in Figure 7 for references to timing of seabreeze.  Also refer to lines 457-460, i.e., "The increase in updraft strength shown by the contours in Figure 7a-b coincide not only with increased winds at BL top and a transition to stronger surface winds (orange square overlaid on BL-average wind speed in Figure 7d) with a southerly component around 18 UTC  (i.e., arrival of the SB), but also bursts in wind speed that are sometimes staggered temporally with updrafts."

There seems to be a confusion between angular frequency and linear frequency which makes the interpretation of observed time scales slightly occluded.  The periodicity in Figure 1b is clearly 2 hours, yet the scalogram is reporting it in Figure 1c as 0.3 hrs., which is probably a factor of 2*pi shorter.  However, the physical process that is affecting the "ripple" in ozone has an obvious time scale of 2 hrs.  This is important because later on the authors divide by 2*pi to make a derivative of the timescales with BLH match what is expected to be the entrainment velocity.

EJS: This is how the wavelet analysis works. It's not a Fourier Transform (FT), which processes an entire data set (or a subset of a data set—SFT) to determine a series of weighted sines and cosines based on the data structure of the signal (a static view).  Rather, the wavelet isolates local changes (i.e., separate peaks/troughs in the data set) that maximize in power if the dilation is chosen to envelope or encompass the width of the spike (dynamic view).  Scaleograms reveal the temporal distribution of data spikes (whether part of sinusoid or not), while plotting dilation on the y–axis gives you a sense of the spectral distribution and possibly the symmetry and shape of the spike.  For instance, note that the "sinusoidal" feature observed in ozone does not have a fixed period.  In fact, the width of the peaks/troughs of the sinusoid changes with time.  The four main peaks in the scaleogram in Figure 1c correspond to a trough, peak, trough, peak in Figure 1b (you can see that they're time–matched on the x–axis), which decreases in width with respect to time.  A slight modification in lines 214–215 aims to resolve this ambiguity:  "….and with respect to wavelet dilation, $\tau$ (dilation--y-axis), to isolate localized data spikes that feature different temporal widths,…."

Specific Comments:

Fig. 1: It is unclear how the periodicity that is so obvious in (b) as ~2 hours, is reported as 0.3 hrs., unless you are reporting the inverse of angular frequency, (Period)/2*pi. This seems important because later on the authors divide by a factor of 2*pi to interpret the time scales changes in a growing ABL as corresponding to an entrainment rate (Eq. 13).

EJS:  Please see my response in the general comments related to periodicity.  We are not isolating sinusoids; we are isolating extrema.  The normalization of 2pi should be clearer with an additional figure (Figure 12) added in Section 4 along with modifications in lines 561-566, i.e., "However, an examination of Figure 12 reveals that $NO_x$ and VOC extrema alternated between troughs and peaks during the BL growth phase, while at the same time exhibiting a reduction in amplitude and a broadening in temporal variability. The temporal spacing observed peak-to-peak (and trough-to-trough) ranged between 2.5 and 5 hours (e.g., can be seen by eyeballing Figure 12d), while $\tau_{max}$ ranged between 0.1 and 1.2 hours, which, if the extremes defining these respective ranges are averaged and divided (i.e., $\frac{\bar{\tau}}{\bar{\tau}_{max}}$), $yields$ a factor close to $2\pi$."

Fig. 1c: Typically wavelet coherence figures include a cone of influence to direct the eye away from the extremes of the figures where the uncertainty in the method is greatest. This might assist the reader in interpreting this and other figures in the manuscript (7-9).

EJS: the cone of influence that I think you are referring is represented by the spectral spreading at a particular time by changing the dilation to retrieve a sort of distribution of power as a function of dilation (e.g., Figure 1d). The shape of the spectral spreading, if identical between different variables (or very close), leads to maximum coherence at that time. That is the point of the MSCM technique. The spectral patterns of variables are mapped out in this way to isolate shared behavior, both with respect to time and with respect to the shape of the spectral spreading. Looking at where the power and shape of the distribution is shared between scaleograms takes care of the concern you are raising, which is already done in the manuscript. There are more quantitative ways that could be adopted in the future to exploit the spectral spreading to understand how shapes of data spikes vary between variables. Currently, however, that is beyond the scope this work.

Fig. 2: Solar noon is more closely tied to PST, not PDT, therefore more like 20 UTC (not 19 UTC as specified in the caption.) Also, while Figs. 2 & 3 are interesting ways to present data, it is very difficult to eyeball a correlation with them. For example, to my eye the (anti) correlation between O3 and NOx/NOy seems stronger than with VOC/NOx. Maybe a simple table with the daytime correlation coefficients for all of these parameters would be helpful to the reader to "calibrate" their eyes. It would help to give some quantified sense of proportion when making statements such as, ""…increased NOx during nights that preceded O3 exceedances." (Line 587)

EJS: Solar noon has been changed to noon for all instances in which solar noon was mentioned in the manuscript. Correlation coefficients have been added for some key variables as shown in Table 1. A few sentences have been added that discuss this table in lines 301-305, which also lines up with the discussion of Figures 2 and 3: "Table 1 summarizes the correlation coefficients for key chemistry and meteorological variables discussed. As can be seen, higher correlations (and anti-correlations) are found in temperature, relative humidity, VOCs, $PM_{2.5}$, $NO_x/NO_y$, and $VOCs/NO_x$ when compared to $O_3$. Variables that are uncorrelated with $O_3$ are surface pressure and BL height. As noted earlier, $O_3$ exceedance periods straddled transitions from high-to-low BL height as well as surface pressure during a limited sampling period of about a month."

Fig. 3: Because there is no ABLH data before 8/10, this work presents an anti-correlation dominated by 2 elevated O3 episodes which occur overlapping the presence of ~3 ABLH minima in the same 20 day time period. This is not a very solid correlation. In fact, what seems more interesting is that the O3 events seem to occur on the "falling edge" of a high ABLH period, that is as the BLH_max is decreasing on the synoptic scale. Nevertheless, the limited time series makes such ideas very limited conjectures.

EJS: I agree that the analysis relied too heavily on Figure 4, which can be problematic as averages can mask statistical distributions. The "falling edge" is an interesting observation, and I concur with your point. Please see lines 289-291 which now says "A transition from deep to shallow daytime BL heights in Figure 3e and a slight reduction in BL-averaged wind speed in Figure 3c coincided with periods of $O_3$

exceedance". Please also note lines 371-380, which discusses Figure 4 as it relates to BL height and surface pressure along with a reexamination of Figure 3.

Figs. 2&3: I think it might be more clear and accurate to run the hour along a slight diagonal as the orthogonal date axis increases (upward to the right). When comparing patterns from left to right (the way we read) it really should be done slightly obliquely in time. But that is just a thought, not a strong recommendation.

EJS: I understand what you are saying, but don't know how that would work. The axes (time and date) are currently orthogonal. There is a discontinuity at 0 and 24 utc where the following day starts directly above the previous day along the y-axis, and where 0 utc is to the left for the following day and 24 utc is the right for the previous day. I'm having a hard time visualizing the plot suggestion. The only plot that comes to mind that preserves temporal continuity is a time series. The point of the plots in their currently form is to understand how the diurnal structure changes during the course of the month as is done here, and to compare with Figure 4, which are diurnal averages. If you give me a plotting example of what you are referring to, then I may consider trying it out.

Line 292: "A large increase in NOx leads to a lowering of VOC:NOx ratio" seems more like a tautology than an interesting point. When the value of the denominator goes up, there typically exists a substantial reduction in the ratio.

EJS: Not if VOCs (the numerator) decrease proportionately or more so than NOx.

Fig. 4: It seems like the main determinant of the high afternoon O3 could easily boil down to which direction the overnight winds are coming from: high O3 is preceded by NE-erly flow that has a lot more VOCs (and potentially many more biogenics from the San Gabriel Mountains) and early a.m. NOx (and PM2.5). This chemical preconditioning gives rise to much greater O3 production throughout the daylight hours.

EJS: This is an excellent point. My chemistry knowledge is somewhat limited, and what you are saying makes a lot of sense. Please see lines 349-351 which says "The northerly winds observed during evenings preceding $O_3$ exceedance events (Figure 4l) may have contributed to increased biogenic VOCs (Figure 4c) advected from the San Gabriel Mountains and increased $PM_{2.5}$ from lingering wildfire smoke".

Fig. 4l: The small difference in afternoon wind direction may be quite significant. The 10-15 degrees greater WDR on low O3 days shows that the Sea Breeze is developing earlier, which is why the T is lower and potentially the advection of precursors has different timing. The longer southerly air is brought to Pasadena during the peak photochemical production hours, the higher the overall O3 will be. Also, just a reminder that simply averaging the numerical wind direction in these plots can be misleading. I am assuming the "average" wind directions are vector averages. Please confirm that is so.

EJS: I completely concur that advection and wind direction is important since the wind will intersect different sources of aerosols based on where winds are coming from. However, it is challenging to comment on such a subtle change, especially since other meteorological quantities stand out compared

to differences in daytime wind direction which were generally minor.  I always average the components before calculating wind direction.

Also, because RH is so strongly dependent on T, I would recommend trying to look at specific humidity or dew point temperature instead of RH.  I suspect it would be the best indicator of the sea breeze that there (aside from lower T).

EJS: I did add dew point to Figure 6 (now Figure 7), but nothing really stands out that suggests a seabreeze passage. Therefore, for seabreeze detection, I rely more on wind direction and speed since that is a clearer indicator for the case examined.  For all other plots, though, I kept relative humidity.

Fig. 4:  It would help to put down the N, number of data points, for each, to get a sense of the statistical power of these comparisons when sampled conditionally against the O3 peak threshold.

EJS:  The exact number of $O_3$ exceedance days is reported in line 287

Line 322: "Increased temperature" should be changed to "increased afternoon temperatures" because the high O3 subsample actually has lower overnight lows.

EJS:  You are correct.  I have made that change

Line 324:  When is wind speed shear reduced?  They seem to vary out of phase quite a bit.  Also, it would be a lot better of a variable, if you are trying to indicate turbulent production, to calculate vector shear (not the wind speed shear):  sqrt[(du/dz)^2 + (dv/dz)^2]

EJS:  You can visually see that the average wind speed shear at night during $O_3$ exceedance is less than non-exceedance nights between the vertical lines.  I'm not sure why the groupings of these different days would lead to wind shear being out of phase.  I have decided to keep wind speed shear because this definition allows me to identify instances of negative shear that develop during the daytime. This is seen particularly when onshore flow develops and a low-level wind maximum well within the BL moves into the Pasadena region. Please see lines 363-364.

Lines 365-371:  The discussion of synoptic details that exist downstream (e.g., tropical cyclone Fred) does not seem all that relevant.  On the other hand, the inverted trough is a common pattern in the warm season across California. This is relevant because there appears to be a lot of wildfire smoke from the north all throughout the region. In fact, the Suomi NPP/VIRS Deep Blue Aerosol Type product shows considerable wildfire smoke in the vicinity of Pasadena.

EJS:  Understood.  I removed the details on Fred but keep other relevant synoptic descriptions. The following has been removed ", with significant flow modifications over the eastern United States as tropical cyclone Fred moves into the Gulf of Mexico"

We are aware of the impact from wildfire smoke.  I believe there were days documented where wildfire smoke advected into the sampling region.  We already addressed the possible impacts from wildfire smoke as indicated by a response to one of the comments made above.

Line 372: I believe that it is very important to be sure that the HRRR output for ABL height is in altitude above ground surface, as opposed to above mean sea level. This should be made clear in the units (m-agl) throughout the manuscript.

EJS: In my experience, ABL output from models is reported in AGL. If you look at the HRRR figure, it should be clear that the ABL is in AGL. For example, Figure 5a clearly shows almost uniform BL heights (<400 m) that extends into the mountain regions. There is no terrain pattern in this field that stands out. I have indicated the ABL is reported in AGL within the Figure 5 caption.

Line 387: You should probably define what a convergence line is exactly. The convergence around Pasadena tends to exist most days with onshore flow because of the San Gabriel Mountains on its northern flank which naturally forces horizontal flow convergence in the presence of southerly wind.

EJS: The point that you are raising was already made clear in lines 393-396 of the manuscript version that you read through.

Fig. 5: It might be more instructive to not fill in the marker identifying Pasadena, so the color scale can be read within the region. How well does your DL estimate of ABL depth compare with the HRRR output in general?

EJS: This paper is not meant to be a model evaluation exercise. However, I did add a BL height comparison for the case study, a curtain plot at Pasadena of wind speed using the HRRR output, and a comparison between component winds before, during, and after the seabreeze transition. This is the new Figure 6. Also, I don't expect to see marked differences between BL heights surrounding the star compared to pixels obstructed by the star, but I did hollow the star which can be seen in the updated Figure 5.

Line 409-410: Can you explain this suggested mechanism? Is there any evidence of a strengthening inversion in this time? In my opinion, there is no sound evidnece of strong subsidence on this day (despite the qualitative arrows annotated in Fig. 6) and there is no reason to believe that increasing static stability increased the winds at that elevation.

EJS: Please see the nearest soundings at Vandenberg AFB and San Diego referenced in the manuscript in lines 460-462. Both show a strong inversion where a drying simultaneously occurs. The large increase in dew point depression is a strong indicator of subsidence. Also, take a look at the new Figure 7d. I added surface pressure. Clearly there is an increase in surface pressure that coincides with the descending wind maximum overnight (first arrow). Interestingly, the increase in pressure coincides with statistics shown in pressure in Figure 4. Lastly, I have added a Figure and a discussion in Appendix C that examines the changes in the profile structure of the descending wind maximum before sunrise (first gray arrow) and as the BL height reaches a maximum (second gray arrow). There is clearly a modification in the winds as the wind maximum displaces downward. The proposed mechanism can be understood in this way: Subsidence can lead to a warming aloft that increases inversion strength. An increase in inversion strength decreases the coupling between residual layer top or BL top as static stability increases, thus minimizing entrainment and allowing for acceleration of the flow to develop.

The downward motion that encounters the top of the boundary results in divergence from above that leads to momentum being transferred horizontally along the direction of background winds. The acceleration of the flow and an increase in wind shear above the inversion could promote shear-induced entrainment, and thus reestablish coupling between the BL and free troposphere. It would be interesting to calculate the Bulk Richardson number across this height, but unfortunately we lack the thermodynamic information to do so.

Lines 424-426: I do not understand this argument. What "oscillations" exactly are you referring to? Second, how/why do they appear related? Which characteristics of the ABL are you referring to and how would those influence the "oscillations"?

EJS: I think the oscillations were clearly described earlier, and while it was unclear what the mechanism behind the oscillations initially was, we did conduct an analysis reserved for Appendix D that describes the intrinsic features of oscillations and the proposed mechanisms. I do modify the text slightly in the main body of the text as well. Please see lines 473-483.

Fig. 6a: The annotated gray arrows indicate a vertical velocity of -200 m/hr = - 5.5 cm/s (day and night). This is a very strong subsidence rate in the lower troposphere, and it is not clear how they were inferred from profiles of wind speed and direction. What makes you confident that the wind is transported downward in such a manner? Momentum is definitely *not* a conservative tracer in the atmosphere. Regardless, the arrow runs from very low wind speed (blue) to strong wind speed (red). Why are these two locations related as indicated by the gray arrow? Furthermore, any observation of a descending scalar from a fixed point measurement is always subject to the potential aliasing by the horizontal advection of a slightly slanted layer. There is no reason to believe those gray arrows, in my opinion.

EJS: Subsidence velocities of this magnitude are not unheard of. A recent study testing different slab models considered similar magnitude subsidence in the same general region, i.e., https://agupubs.onlinelibrary.wiley.com/doi/full/10.1029/2020JD033775. Please see my comments earlier related to supporting plots. Also, look at the new figure I added showing wind speed profiles spanning the first and second gray arrow (Figure C1). A discussion related to this plot can be found in lines in Appendix C. Also, I don't see where it is indicated within the manuscript that these two events are related. I'm simply pointing out patterns observed within the DL data and discussing them qualitatively. Your comment on advection is well taken.

Fig. 6c: This is a very unusual Doppler lidar diurnal vertical velocity plot. Typically there are domains of updrafts and downdrafts intercalated every 5-10 minutes throughout the ABL (e.g., Lothon et al., 2009; Maurer et al., 2016). There appears to be no downdrafts observed anywhere in this plot, which violates mass continuity. Furthermore, since the DL measurement is an average fo 11.5 minutes every 15 minutes, there is a chance of aliasing higher frequency components into this dataset. This brings up the fact that it would be reassuring to see some of the data from the DL prior to getting handed over to the more

complex mathematical treatments.  For example, how do the w-variance profiles compare with the literature, and the boundary layer heights compare to HRRR, etc.

EJS:  The scan cycle is to blame for this.  Please refer to Section 2.1.1 detailing the stationary lidar scanning cycle.  The winds were measured over a 11.5-minute period and averaged.  The individual updrafts and downdrafts get smeared, while stronger upward motions with longer temporal durations are revealed as the seabreeze moves in with a near surface wind maximum, effectively undercutting the overlying atmosphere and promoting mechanical lift, and, if the relatively cooler air propagates over a heated surface, an enhancement in near-surface buoyancy fluxes.  We do not rely heavily on this plot and only focus on the strongest updrafts observed by the lidar.  Furthermore, Figure 6 (now Figure 7) has been modified.  Curtain plots of vertical velocity is no longer included.

Line 464: "…provided that the velocity of overturning eddies does not change appreciably."  But quite the contrary: it absolutely does!  The convective velocity scale is going to increase with increasing surface buoyancy fluxes throughout the day, and while the BLH will also, the former increases at a power of 1/3.  Thus the large eddy turnover time scale will be proportional to BLH^(2/3).  Using a simple slab model convective boundary layer model (e.g. CLASS, https://classmodel.github.io/) one finds that this time scale increases monotonically over the course of the daytime heating (from ~5 to ~20 minutes).

EJS:  I understand and completely agree.  In hindsight, I realize that this was a dangerous thing to say and could have been left out of the sentence altogether.  I was going back and forth between the BL modeling analysis and writing this section, and believe that I merged the "nearly constant growth rate of the BL height" with this part of the discussion.  I will remove this part of the sentence and ensure elsewhere reflects this change as a precaution. Note that references to velocity scales have been changed to BL growth rate.

The Doppler Lidar data should allow for an estimate of w* by convective similarity, and BLH, therefore the large eddy turnover time should be able to be estimated:  tau = BLH/w* from the measurements directly.

EJS:  I think the language that I used in the manuscript related to this part of the discussion has led to the impression that I'm attempting to derive a convective velocity scale, $w_*$.  Clearly there is a conflation between what I'm defining as the change in the boundary layer height with respect to time and a velocity scale.  I call the change in BL height a velocity scale, but, in reality, that is not the correct language that should be used since velocity scale is really more appropriate in defining the convective thermals that develop within a boundary layer that also change with time as result of increased surface buoyancy flux.  Furthermore, there are no assumptions about the form of the time-scales.  This is a data-driven exercise.  Therefore, we have elected not to expand beyond the current analysis and focus only on interpreting the fine-scale features of measurements in the context of BL growth and dynamics according to the measurements.

Line 466:  There is so much turning of the winds throughout this day that you are observing many different things affecting your time series via simple advection differences. This is the crux of the problem with the interpretation of all these "covariances". Differential advection is likely very dominant in this system.

EJS: There is a gradual turning of winds during this time period. The time variations broaden with respect to time as the BL height grows. This is highlighted in Figures 8 through 10. However, I do agree that advection needs to be mentioned as a caveat and perhaps as a way to encourage using a network of remote sensing instruments, not just one instrument. This is a challenge when only one measurement platform is available. We do add an additional plot that highlights the time variations within a time series for $NO_x$ and VOCs (Figure 12). There are clear troughs and peaks that decrease in amplitude at the same time they broaden (longer period time variations). The trough-peak that happens in sequence within this plot is used to justify the 2pi normalization as already mentioned above, and the reduction in amplitude and increase in period of temporal variations occurs as the BL deepens. The veering in wind direction across the BL is gradual with winds generally remaining weak within the BL.

Lines 521-523: Again, this discussion entirely ignores the time scales of horizontal advection and the veering wind which brings in different concentrations which is likely contributing to the variations in the chemical species significantly. Furthermore, when you bring up processes like mixing changing concentrations which change chemical reaction rates, you are blending the transport and chemical reaction terms (all of which are going to have a wide range of time scales: from 10 minute for a reactive VOC and NO2, to half a day for less reactive VOCs).

EJS: This is acknowledged in several places within the manuscript. Please see lines 482-483, 593-594, and the wind direction analysis in section 5 that addresses this concern.

Line 545: On Aug 16 it looks like the SB did not really fully influence the sampling site until 20-21 UTC (nearly outside of the subdomain you are studying here: 14-20 UTC). I believe it is critical to mark the arrival of the marine layer at Pasadena, and it will likely be most apparent when looking at dew point temperature or specific humidity in conjunction with the other variables (e.g. Fig. 6).

EJS: We did plot dew point and specific humidity to check this. There are no clear perturbations in these fields that stand out (please see updated Figure 7d). The response in the wind field seems like a more reliable indication for this day. Note that in Figure 7d there is a jump in wind speed around 18 utc with sustained strength. This occurs as winds transition to more southerly from easterly-southeasterly. There is a strong onshore component with shorter fetch from the coastal ocean into Pasadena. The HRRR also shows more onshore flow by 19 utc (Figure 5 and 6), which supports the timing of a wind speed shift. Therefore, we argue using changes in the winds over a change in the dew point as the time where onshore flow begins to impact Pasadena, at least for this case. The orange square in Figure 7d represents the SB arrival time.

Line 549: Where does this 17 UTC time come from? The August 16 case study indicates the SB arrival time is more like 19-21 UTC (Fig. 6, based on wind direction veering to southwesterly, the direction of the nearest coastline, and the premature fall of the air temperature).

EJS:  The forcing and the background conditions that promoted onshore flow were not uniform.  The timing of onshore flow was variable from day-to-day.  We have removed Figures 11 and 12 from the original manuscript and replaced those figures with Figure 13, which consolidates Figures 11 and 12 while adding wind direction information.  This allowed for a more intuitive explanation and a clearer discussion as evident in major changes in Section 5.

Line 551-553:  This discussion seems very speculative. For instance, to eliminate the other variable pairings is to assume that n=2 is a decent account of how they 'normally' behave.  Further, the subjective grouping of "high frequency" and SB arrival is extremely fuzzy.  When does the SB arrive on each day (Aug 16 it looked more like 20 UTC), and what is high frequency?  There are plenty of scatter points that are below 1 hr in period.

EJS:  This is an objective method.  We are simply identifying the spectral similarity between variables as outlined in Appendix A.  Essentially, we identify variables that share a similar temporal evolution.  We limited our analysis to n=2, because while there were spectral similarities with strong overlap for 3 variables, much could be said with just looking at one variable or a pair of variables.

Lines 554-556:  This exercise surrounding Figure 12 seems fraught.  BLH and time are going to be strongly correlated in this time interval (in fact, monotonically linked).  So these figures (Fig 12) look a lot like the previous set (Fig. 11) just rotated around the x=y line. And the selection of the subset in the red circles seems arbitrary as they do not visually cluster in any noticeable way.

EJS:  Please see the updated discussion in Section 5 with a new Figure (Figure 13) that consolidates Figure 11 and 12 in addition to including wind direction.

Line 563-566:  It seems unlikely that the SB arrives in Pasadena by 8-10 a.m.  You can look for yourself with the wind direction, specific humidity, etc. shifts daily.  But even so, you are saying you recognize that everything that can affect high frequency changes in a reactive scalar like O3 could be happening.  That is true in the most general sense.  What type of "dynamical interactions" and "precursor reactions" are being referred to? It might be instructive to inspect the scalar budget equation of these reactive compounds.

EJS:  I now say onshore flow.  There were days where a southerly-to-southwesterly flow developed early.  This could be a result of ideal forcing conditions promoting the arrival of onshore flow earlier.

Lines 569-570: Why present this data if any associations that a reader is inclined to infer from the figures is always going to be statistically insignificant? I would recommend leaving these small event counts out of your analysis altogether. They are misleading, in my opinion.

EJS:  There are plenty of scatter points when analyzing single variables, so we will keep this analysis.  We understand the limitations of the time record, and we no longer highlight

clusters of scatter points within the plots as done before because we have consolidated the plot which now has more information including wind direction. This can be viewed in Figure 13 and the discussion of this figure in Section 5. We have also quantified the scatter distributions for single variables which is shown in Figure 14 and discussed in Section 5.

Line 578: I would avoid the use of the word "stable" because of its preeminence in buoyancy/mixing.  If what you mean is "stationary" (i.e., not time dependent) I would use that term instead.

EJS:  I have inserted "chemically" before "stable"

Line 594: I believe it is very unlikely that the difference of 0.5 m/s at the surface is going to influence BLH.  August in SoCal will not typically produce neutral ABLs, they tend to be strongly convective. Surface shear production is not the dominant source of turbulent kinetic energy. Without knowing what the subsidence difference is between low and high O3 days, you cannot suggest that this is a reason the ABL top is lower.  The differences in the strength and timing of the sea breeze, which brings lower T air into the region (and lower BLH), is much more likely to be the cause of these modest differences.

EJS:  The statement in line 594 is no longer in the manuscript based on changes made within the manuscript.

Lines 597-599:  The greater winds and deeper nocturnal ABLs would lead to increased dry deposition of O3 and NO2 in a thicker layer overnight.  This does not necessarily reduce the role of titration, but reduces the next day's Ox levels. Also the daytime BL heights have very little to do with the ~50 m difference in their initial morning values (Driedonks, 1982).

EJS:  The conclusion section has been significantly adjusted which should address this comment.

Line 603:  At what elevation is the "observed" wind shift you are referring to?  One can find a wind shift in that figure at some elevation just about any time of day.

EJS:  Please see the changes in lines 674-675, "An interesting meteorological feature worth noting was the semi-diurnal pressure pattern, whose troughs lined up near transitional periods (sunset and sunrise)."

Line 608-612:  The winds are not advected around by the flow in a conservative manner like a non-reactive scalar is.  They are strongly influenced by several thermal circulations in this region (varying pressure gradients with height) that are all changing strength throughout the day (upslope southerly flow in the a.m. and southwesterly sea breeze flow later in the day.)  "Patterns of descent" suggested by a wind pattern is highly speculative without interrogating the entire Navier-Stokes equation (and also importantly the thermal wind.)

EJS:  Please see supporting plots referenced in the text and accompanying description in lines 445-455, and discussion and analysis of Figure C1 in Appendix C.  I disagree that the

full suite of equations would have to be investigated.  The thermal wind would be a good diagnostic for evaluating the vertical wind shear with respect to the horizontal thermal gradient at a constant pressure, but is viewed as unnecessary as that would only infer the mechanism behind the vertical wind structure.  In appendix C, we employ mass continuity to come up with a back-of-the-envelope calculation of horizontal propagation, but stop short of analyzing thermodynamic responses or influences since we lack this information.

Line 613:  The region of higher wind speeds (>5 m/s) above the ABL (~600-1500 m) is more or less continuous throughout the day from the southeast.  There is only one period of ~ 1hr near solar noon when the winds accelerate to 8-9 m/s.

EJS:  Please refer to Figure C1 and discussion therein.

Line 614:  Bear in mind that one does not need to hypothesize a temporary, thin wind jet atop the ABL to "initiate" entrainment.  Entrainment is sure to be occurring vigorously throughout the day because of strong surface heat fluxes in SoCal in August.

EJS:  Yes, but the BL rarely exceeded 1.5 km in the area studied, and often a strong inversion was observed (evident in soundings and ACARS profiles that we looked at). While buoyancy is the primary driver of BL entrainment in atmospheric convection situations, the departure from idealized convective BL conditions challenges the simplified ideal view.  The flux profile is no doubt significantly modified, which would yield changes with respect to time of the flux ratio between the surface and the inversion (i.e., beta would not be constant).

Line 628:  The NOx and VOC concentrations do not increase, but rather their variance does.

EJS:  Please see lines 707-709, i.e., "The temporal widths of extrema in $NO_x$ and VOCs increased as the BL deepened, which is corroborated well with variations in the BL structure with respect to time that ranged from 15 minutes shortly after sunrise to 1.5 hours as the BL climaxed"

Line 628:  I think it is better to be more specific with the wording here:  it is not any other "structure" than the ABL height, correct? If not, then specify what "structure" parameters you are referring to.

EJS:  Have changed to "height"
* * *
This manuscript aims to understand meteorological and chemical variability associated with boundary layer growth and sea breezes. The authors use a novel statistical approach to understand relationships between the observed variability. The topic is suitable for publication to ACP. Two major comments and a few minor comments are below.

Major comments:

1) The paper does not mention nighttime land breezes, but Figure 4l suggests that nighttime land breezes occurred prior to high ozone days, which probably played a role in poor air quality. A discussion on these land breezes on the observed air quality should be worked into the manuscript. See more pertaining to this issue below.

EJS: Thank you for your comment. I have expanded the discussion on northerly flows in several areas of the paper which can be found in the following lines:

Lines 343-345: "While little can be ascertained from the BL-averaged vertical velocity in Figure 4j, the wind direction in Figure 4l was northerly (southwesterly) during nights preceding days where $O_3>=70$ ppb ($O_3<70$ ppb) before converging to a southwesterly wind into the afternoon hours in support of onshore flow."

Lines 349-351: The northerly winds observed during evenings preceding $O_3$ exceedance events (Figure 4l) may have contributed to increased biogenic VOCs (Figure 4c) advected from the San Gabriel Mountains and increased $PM_{2.5}$ from lingering wildfire smoke.

The lines above point out the differences in wind direction while noting the possibility of enhanced VOCs from biogenic and biomass burning sources. We stop short of calling a northerly flow a land-breeze because of the discussion given in the lines below and arguments made below the lines referenced:

Lines 441-445: "A shallow BL developed during the evening (less than 60 m) with weak easterly winds within the first 500 m that were occasionally interspersed with shallow northerly flows extending across the first 100 m from the surface. Above 500 m, winds increased and veered northwesterly, likely as a result of combined influences of clockwise flow associated with the offshore high-pressure system and counter-clockwise flow from the inverted trough converging over the San Gabriel Mountains to the north."

Above we note the northerly flows in support of synoptic and large mesoscale conditions, which make it difficult to tie what is observed at Pasadena to a land breeze. Furthermore, because our observations are in Pasadena against the San Gabriel Mountains 50 km away from the coast, we stop short of calling what is observed a land breeze or offshore flow. What we did notice was a descending wind into Pasadena that we discuss more detail within an added appendix (Appendix C). The points made in the lines above are reiterated in the conclusions section in addition to a discussion in Section 5. Section 5 has been significantly changed based on comments from a separate reviewer. We have consolidated the old Figures 11 and 12 into a single figure (Figure 13), which contains scatter color-coded by wind direction. We have also added Figure 14 to examine the statistics more quantitatively in this section based on your recommendation below.

2) While the scaleogram technique is a novel way of investigating the variability of meteorological and chemical variability within the atmosphere (Figures 7-9), it is not clear if

combining the maxima of the spectral peaks from the scaleograms during these two different meteorological processes (boundary layer growth and sea breezes) results in statistically significant relationships on variability and boundary layer height (Figures 10-12). The analysis and results associated with Figures 10, 11, and 12 might show more statistically significant results if you only look at the impact of boundary layer growth by not including data points once a SB arrives. At least that is my hypothesis.

EJS: The reason why we chose not to separate the daytime growth of the BL from the arrival of a SB was because the day-to-day differences in wind direction shifts, wind speed intensification, and response in in situ measurements. The timing of the SB was not identical each day. Furthermore, we wanted to isolate the portion of the BL that exhibited growth, whether contaminated with a SB or not, to examine the fine structure variability as observed by in situ measurements with respect to increased BL depth. The differences in time of the arrival of the SB in Figures 11 and 12 (now Figure 13) led us to not isolate and remove the portion of BL growth contaminated with a SB. Therefore, we have maintained our position to examine the entire BL growth phase, noting the challenges of disentangling the SB and daytime heating effects on the time evolution of the BL height.

Other comments:

Page 1, line 14: Briefly state what the findings from the cast study are.

EJS: I significantly modified the abstract to satisfy this comment. Please see the text below, which is now assimilated into the abstract shown in the updated manuscript.

"Separate analyses are dedicated to differentiating the synoptic conditions during $O_3$ exceedance (>70 ppb) and non-exceedance (<70 ppb) days, and the fine structure variability of in situ chemistry measurements during BL growth and seabreeze (SB) transitions.

Diurnal analyses spanning August 2021 revealed a markedly different wind direction during evenings preceding $O_3$ exceedance (northerly) versus non-exceedance (easterly) days. Increased $O_3$ occurred simultaneously with warmer and drier conditions, a reduction in winds, and an increase volatile organic compounds (VOCs) and fine particulate matter ($PM_{2.5}$). While the average BL height was lower and surface pressure was higher, the day-to-day variability of these quantities led to an overall weak statistical relationship. Investigations focused on the fine structure variability of in situ chemistry measurements superimposed on background trends were conducted using a novel Multivariate Spectral Coherence Mapping (MSCM) technique that combined the spectral structure of two or more independent measurements through a wavelet analysis as reported by maximum-normalized scaleograms. A case study was chosen to illustrate the MSCM technique, where the dominant peaks in scaleograms were identified and compared to BL height during the growth phase. The temporal widths of peaks ($tau\_\{max\}$) derived from VOC and nitrogen oxide ($NO_x$) scaleograms, and scaleograms combining VOCs, $NO_x$, and variations in BL height indicated a broadening with respect to time time as the BL increased in depth. A separate section focused on comparisons between $tau\_\{max\}$ and BL height during August 2021 revealed uncorrelated or weakly correlated scatter, except in the case of VOCs when really large $tau\_\{max\}$ and relatively deep BL heights were ignored. Instances of large

tau_{max} and increased BL height toward a maximum occurred near sunrise and as onshore flow entered Pasadena, respectively. Wind transitions influence both the dynamical evolution of the BL and tracer advection, and thus offering additional challenges when separating factors that influence the fine structure. Other insights gained from this work include observations of descending wind jets from the San Gabriel Mountains that were not resolved by the HRRR model, and the derivation of intrinsic properties of oscillations observed in $NO_x$ and $O_3$ during the interaction between a SB and enhanced winds above the BL that flowed in opposition to the SB. "

Page 4, line 4: Capitalize the first letter in this sentence.

EJS: The "in" in "in situ" is now capitalized.

Page 5, line 23: Re-word "accessible online at S. (2021)."

EJS: Now says "..accessible online (Brown 2021)."

Page 15, line 327, Figure 4l, and part of the discussion in the conclusion: The nighttime northerly winds is likely a sign that a land breeze has formed. Including a discussion on the land breeze in the paper would be beneficial. This probably plays a large role in the high ozone days. It results in air pollution to recirculate and stick around in the LA Basin until synoptic scale winds are strong enough to push them over the mountains. The nighttime southwesterly nighttime winds preceding low ozone days suggest pollution is being transported over the mountains. Does the HRRR simulate a land breeze at night? Figure 5 does not cover nighttime hours.

EJS: Please see the response to major comment 1. We have expanded the discussion of northerly flows during evenings preceding days where $O_3$ became elevated, but stopped short of calling this a land-breeze given the position of Pasadena relative to the coast and because the northerly flows appeared to originate from elevated terrain or across elevated terrain. For these reasons, we do not call this a land-breeze or offshore flow. Also, we were not able to determine that flows moved across elevated terrain during instances of onshore flow. This would depend on the background synoptic conditions in addition to differential thermal forcing across the coast that drives the SB. It's also difficult to infer that onshore flow surmounted elevated terrain in Figure 5.

Figure 5: Consider showing a nighttime hour plot. Also add observed surface wind barbs to the figures.

EJS: We have replaced the 8am plot with a 6am plot, which is early enough to capture flows directed towards offshore. We also looked at earlier times as well, but that did not reveal additional information. We have declined to add the wind barb observation on these maps because it is a single data point and because we compare the winds between the HRRR and the lidar with our new Figure 6.

Page 18, line 407: change "BL height, and" to "BL height growth, and"

EJS: Change has been adopted

Page 18, line 417: In addition to entrainment, also detrainment. Consider also mentioning detrainment or BL-free troposphere exchange.

EJS: We have modified this sentence slightly in the following way in lines 455-457: "…which can limit entrainment/detrainment between the BL and free troposphere." We removed "coupling between" as entrainment/detrainment infers coupling already.

Page 18, lines 418-420: Why do some pulses cause an increase in NOx and decrease in O3 and some cause the opposite?

EJS: This is challenging to address because we don't have vertical profile information about $O_3$ or $NO_x$.  For this reason, we cannot comment on the exact nature of why these variables are 180 out of phase.  We looked at $O_x$, which exhibited smaller amplitude variations compared to $O_3$ and $NO_x$.  We are confident that what is observed is dynamically driven.  In the conclusions, we say the following "A phase shift of 180 degrees observed between $O_3$ and $NO_x$ oscillations during the SB is less clear.  However, if NO$_{x}$ is somehow replenished by a nearby source or the reactivity time-scales of NO$_{x}$ are considerably longer than dynamical time-scales, then the variations would be dominated by the transport dynamics. Under these assumptions, the 180-degree difference between $O_3$ and $NO_x$ would therefore be related to an opposing profile structure (one quantity increases with height while the other decreases with height) subjected to the same dynamics that lead to mixing via turbulent eddies." In lines 715-720.

Page 20, line 456. Change "NOx, or VOCs-Ox did" to "NOx, or VOCs. Ox pairings did"

EJS: I have replaced the hyphen with a semicolon.

Figure 10: The NOx lines may look similar to the others if you don't include data points dealing with the SB. Consider only using data points before the SB moves over the measurement location.

EJS: Please see the response to major comment 2.  We decided not to remove the portion of the BL growth contaminated by the SB because the arrival of the SB varied day-to-day, and because we wanted to isolate the full time period encompassing BL growth even if the BL growth phase encompassed the arrival of the SB.  Furthermore, truncating the BL growth phase would reduce the scatter points used for fits, which is undesirable.

Page 26, lines 541-553: Quantify. Hard to see a trend when looking at any of these figures.